# The SARS-CoV-2 Spike protein has a broad tropism for mammalian ACE2 proteins

Carina Conceicao[1], Nazia Thakur[1], Stacey Human[1], James T. Kelly[1], Leanne Logan[1], Dagmara Bialy[1], Sushant Bhat[1], Phoebe Stevenson-Leggett[1], Adrian K. Zagrajek[1], Philippa Hollinghurst[1,2], Michal Varga[1], Christina Tsirigoti[1], Matthew Tully[1], Chris Chiu[1], Katy Moffat[1], Adrian Paul Silesian[1], John A. Hammond[1], Helena J. Maier[1], Erica Bickerton[1], Holly Shelton[1], Isabelle Dietrich[1], Stephen C. Graham[3], Dalan Bailey[1]*

1 The Pirbright Institute, Woking, Surrey, United Kingdom, 2 Department of Microbial Sciences, Faculty of Health and Medical Sciences, University of Surrey, Guildford, United Kingdom, 3 Department of Pathology, University of Cambridge, Cambridge, United Kingdom

☯ These authors contributed equally to this work.
* dalan.bailey@pirbright.ac.uk

**Data Availability Statement:** All relevant data are within the paper and its Supporting Information files.

**Funding:** This work was supported by the following grants to DB: a UK Research and Innovation (UKRI;

## Abstract

SARS Coronavirus 2 (SARS-CoV-2) emerged in late 2019, leading to the Coronavirus Disease 2019 (COVID-19) pandemic that continues to cause significant global mortality in human populations. Given its sequence similarity to SARS-CoV, as well as related coronaviruses circulating in bats, SARS-CoV-2 is thought to have originated in Chiroptera species in China. However, whether the virus spread directly to humans or through an intermediate host is currently unclear, as is the potential for this virus to infect companion animals, livestock, and wildlife that could act as viral reservoirs. Using a combination of surrogate entry assays and live virus, we demonstrate that, in addition to human angiotensin-converting enzyme 2 (ACE2), the Spike glycoprotein of SARS-CoV-2 has a broad host tropism for mammalian ACE2 receptors, despite divergence in the amino acids at the Spike receptor binding site on these proteins. Of the 22 different hosts we investigated, ACE2 proteins from dog, cat, and cattle were the most permissive to SARS-CoV-2, while bat and bird ACE2 proteins were the least efficiently used receptors. The absence of a significant tropism for any of the 3 genetically distinct bat ACE2 proteins we examined indicates that SARS-CoV-2 receptor usage likely shifted during zoonotic transmission from bats into people, possibly in an intermediate reservoir. Comparison of SARS-CoV-2 receptor usage to the related coronaviruses SARS-CoV and RaTG13 identified distinct tropisms, with the 2 human viruses being more closely aligned. Finally, using bioinformatics, structural data, and targeted mutagenesis, we identified amino acid residues within the Spike–ACE2 interface, which may have played a pivotal role in the emergence of SARS-CoV-2 in humans. The apparently broad tropism of SARS-CoV-2 at the point of viral entry confirms the potential risk of infection to a wide range of companion animals, livestock, and wildlife.

www.ukri.org) Medical Research Council (MRC) New Investigator Research Grant (MR/P021735/1), a UKRI Biotechnology and Biological Sciences Research Council (BBSRC, www.ukri.org) project grant (BB/R019843/1) and Institute Strategic Programme Grant (ISPG) to The Pirbright Institute (BBS/E/I/00007034, BBS/E/I/00007030 and BBS/E/I/00007039) and an Innovate UK Department for Health and Social Care project (SBRI Vaccines for Global Epidemics – 795 Clinical; Contract 971555 'A Nipah vaccine to eliminate porcine reservoirs and safeguard human health'). SCG is a Sir Henry Dale Fellow, jointly funded by the Wellcome Trust and the Royal Society (098406/Z/12/B). The funders had no role in study design, data collection and analysis, decision to publish, or preparation of the manuscript.

**Competing interests:** The authors have declared that no competing interests exist.

**Abbreviations:** ACE2, angiotensin-converting enzyme 2; APN, aminopeptidase N; BHK-21, baby hamster kidney; BVDV, bovine viral diarrhoea virus; -COVID-19, coronavirus disease 2019; COOT, Crystallographic Object-Oriented Toolkit; CSV, comma-separated values; Ct, cycle threshold; DF-1, chicken embryonic fibroblast cell line; DL, detection limit; DPP4, dipeptidyl peptidase 4; DMEM, Dulbecco's Modified Eagle Medium; EM, electron microscopy; FBS, foetal bovine serum; FMDV, foot and mouth disease virus; FSC-A, forward scatter area; GFP, green fluorescent protein; HA, human influenza hemagglutinin tag; HG3, Hazard group 3; MAFFT, Multiple Alignment using Fast Fourier Transform; MERS-CoV, Middle East respiratory syndrome-related coronavirus; MEM, minimum essential medium; MOI, multiplicity of infection; NE, non-enveloped; PEI, polyethyleneimine; PBS T, PBS-Tween; PDB, Protein Data Bank; PE, Phycoerythrin; PHE, Public Health England; PSI, position-specific iterative; qPCR, quantitative PCR; RaTG13 pps, RaTG13 pseudoparticles; RBD, receptor binding domain; RLU, relative light units; SARS-CoV, SARS Coronavirus; SARS-CoV pps, SARS-CoV pseudoparticles; SARS-CoV-2, SARS Coronavirus 2; SARS-CoV-2 pps, SARS-CoV-2 pseudoparticles; SSM, secondary structure mapping; TMPRSS2, transmembrane protease serine 2; TPCK, Tosyl phenylalanyl chloromethyl ketone; USB, universal serial bus; WT, wildtype.

## Introduction

The β-coronavirus SARS Coronavirus 2 (SARS-CoV-2) emerged in late 2019, causing a large epidemic of respiratory disease in the Hubei Province of China, centred in the city of Wuhan [1]. Subsequent international spread has led to an ongoing global pandemic, currently responsible for over 43 million infections and 1,100,000 deaths (as of October 26, 2020, Johns Hopkins University statistics; https://coronavirus.jhu.edu/map.html). As for SARS-CoV, which emerged in China in late 2002, and Middle East Respiratory Syndrome-related Coronavirus (MERS-CoV), which emerged in Saudi Arabia in 2012, the original animal reservoir of zoonotic coronaviruses is thought to be bats [2]. Spillover into humans is suspected or proven to be facilitated through an intermediate host, e.g., civets for SARS Coronavirus (SARS-CoV) [2] or camels for MERS-CoV [3]. For SARS-CoV-2, a bat origin is supported by the 2013 identification of a related coronavirus RaTG13 from *Rhinolophus affinis* (intermediate horseshoe bat), which is 96% identical at the genome level to SARS-CoV-2 [1]. Identifying the animal reservoir of SARS-CoV-2, and any intermediate hosts via which the virus ultimately spread to humans, may help to understand how, where, and when this virus spilled over into people. This information could be vital in identifying future risk and preventing subsequent outbreaks of both related and unrelated viruses. Concurrent to this, there is also a need to understand the broader host tropism of SARS-CoV-2 beyond its established human host, in order to forewarn or prevent so-called reverse zoonoses, e.g., the infection of livestock or companion animals. The latter could have serious implications for disease control in humans and consequently impact on animal health and food security as we seek to control the Coronavirus Disease 2019 (COVID-19) pandemic.

The process of viral transmission is complex and governed by a range of factors that in combination determines the likelihood of successful infection and onward spread. The first barrier that viruses must overcome to infect a new host, whether that be typical (of the same species as the currently infected host) or atypical (a new species), is entry into the host cell. Entry is governed by 2 opposing variables: the first being efficient virus binding to the host cell and the second being host-mediated inhibition of this process, e.g., through virus-specific neutralising antibodies. In the case of SARS-CoV-2, it is likely that in late 2019, the entire global population was immunologically naïve to this virus, although there is debate as to whether pre-existing immunity to the endemic human-tropic coronaviruses, e.g., OC43 and HKU1, provides any cross-protective antibodies to help mitigate disease symptoms [4]. To compound this, the rapid global spread of SARS-CoV-2, combined with emerging molecular data [5,6], has clearly demonstrated that SARS-CoV-2 is efficient at binding to and entering human cells. However, how widely this host range or receptor tropism extends and the molecular factors defining atypical transmission to nonhuman hosts remain the subject of intense investigation.

Coronavirus entry into host cells is initiated by direct protein–protein interactions between the virally encoded homo-trimeric Spike protein, a class I transmembrane fusion protein found embedded in the virion envelope, and proteinaceous receptors or sugars on the surface of host cells [7]. The high molecular similarity of β-coronaviruses allowed the rapid identification of angiotensin-converting enzyme 2 (ACE2) as the proteinaceous receptor for SARS-CoV-2 [8,9], and structural studies characterising Spike and its interactions with ACE2 have quickly followed [5,6,10,11] and extended recently to include RaTG13 [12]. These studies have identified a high affinity interaction between the receptor binding domain (RBD) of Spike and the N-terminal peptidase domain of ACE2, which for SARS-CoV was shown to determine the potential for cross-species infection and, ultimately, pathogenesis [13].

The availability of ACE2 gene sequences from a range of animal species enables study of the receptor tropism of β-coronavirus Spike proteins. This can be achieved through computational

predictions based on ACE2 sequence conservation [14] or, more directly, with functional experimental investigation [1]. In this paper, we examined whether ACE2 from 22 different species of livestock, companion animals, and/or wildlife, alongside human ACE2, could support the entry of SARS-CoV-2, SARS-CoV, and RaTG13. Using 2 distinct assays, we identified that SARS-CoV-2 has a broad receptor tropism for mammalian ACE2 proteins including human, hamster, pig, and rabbit. Efficient infection via these ACE2 receptors was subsequently confirmed using live SARS-CoV-2 virus. Interestingly, the host range of SARS-CoV-2 was similar to that of SARS-CoV, yet distinct from its closest genetic relative RaTG13. The genetic determinants of this receptor tropism were further investigated, and individual amino acids within the Spike–ACE2 interface were identified as critical to β-coronavirus human tropism. This research has identified vertebrate species where cell entry is most efficient, allowing prioritisation of in vivo challenge studies to assess disease susceptibility. Combining this with increased surveillance and improved molecular diagnostics could help to prevent future reverse zoonoses.

## Results

### The SARS-CoV-2 binding site on ACE2 is highly variable

Recent structural and functional data have shown that SARS-CoV, SARS-CoV-2, and other β-coronavirus (lineage B clade 1) Spike proteins bind the same domain in ACE2 to initiate viral entry [5,6,8–10]. We thus hypothesised that SARS-CoV-2 could use the ACE2 receptor to infect a range of nonhuman, non-bat hosts. To this end, we synthesised expression constructs for human ACE2 as well as orthologues from 22 other vertebrate species, including 9 companion animals (dogs, cats, rabbits, guinea pigs, hamsters, horses, rats, ferrets, and chinchilla), 7 livestock species (chickens, cattle, sheep, goats, pigs, turkeys, and buffalo), 4 bat species (horseshoe bat, fruit bat, little brown bat, and flying fox bat), and 2 species confirmed or suspected to be associated with coronavirus outbreaks (civet and pangolin). There is 62% to 99% sequence identity between these proteins at the amino acid level (76% to 99% when excluding the 2 bird sequences), and their phylogenetic relationships are largely consistent with vertebrate phylogeny, although the guinea pig sequence was more divergent than predicted (Fig 1A). Examining the conservation of amino acids at the SARS-CoV-2 binding site on the surface of the ACE2 protein revealed a high degree of variation across mammalian taxa (Fig 1B and 1C), suggesting that SARS-CoV-2 receptor binding may vary between potential hosts. This variation was also evident when aligning the 23 ACE2 sequences included in our study, which identified a number of highly variable residues within the overlapping SARS-CoV and SARS-CoV-2 binding sites, including Q24, D30, K31, H34, L79, and G354 (Fig 1D). Our first step was to ensure efficient and equivalent surface expression of these ACE2 proteins on target cells. To this end, their N-terminal signal peptides were replaced with a single sequence from the commercially available pDISPLAY (Thermo Fisher Scientific, United States of America) construct (Fig 1E). In addition, the ectodomain was fused with an HA-epitope tag to allow the specific detection of surface expressed protein. Western blot of whole cell lysates together with flow cytometric analysis of cell surface expression confirmed that in the majority of cases, the 23 ACE2 proteins were expressed to similar levels, thereby allowing side-by-side comparison of their usage by SARS-CoV-2 (Figs 1F and 1G and S1). The marked exceptions were flying fox bat and guinea pig ACE2 where protein expression and cell-surface presentation were barely detectable (Fig 1F and 1G). The cause of this poor expression is unknown, potentially arising due to errors in the ACE2 sequences available for these species (see Methods; Phylogenetic analysis). Since the available sequence accuracy for these 2 genes would need to be explored further, these 2 ACE2 proteins were excluded from our subsequent experiments.

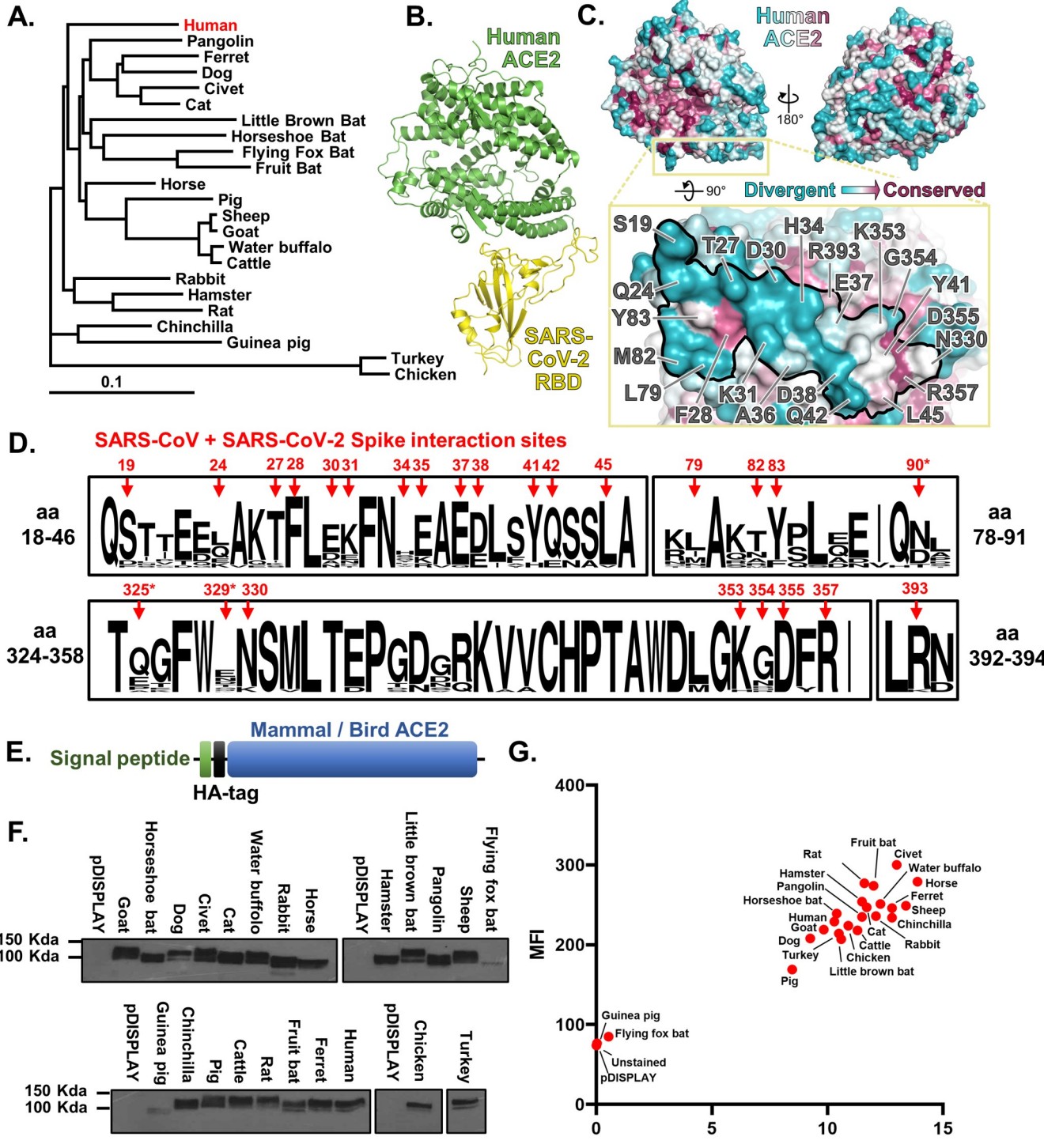

**Fig 1. The SARS-CoV-2 binding site on ACE2 is highly variable.** (A) A phylogenetic tree of ACE2 proteins assembled using the neighbor-joining method [51] conducted in MEGA7 (Temple University, USA) [52] with ambiguous positions removed. The tree is drawn to scale, and support was provided with 500 bootstraps. (B) Structure of human ACE2 ectodomain (green) in complex with the RBD of SARS-CoV-2 [10]. (C) Conservation of mammalian ACE2 amino acid residues, estimated from site-specific evolutionary rates [50], mapped onto the surface of the ACE2 ectodomain [10], and coloured from blue (divergent) to purple (conserved) and presented in 2 orientations. Inset depicts the SARS-CoV-2 binding region of ACE2 (outlined), with residues that contact the SARS-CoV-2 RBD highlighted [6]. (D) WebLogo (University of California, Berkeley, USA) [53] plots summarising the amino acid divergence within the mammalian and bird ACE2 sequences characterised in this study. The single letter amino acid (aa) code is used with the vertical height of the amino acid

representing its prevalence at each position in the polypeptide (aa 18–46, 78–91, 324–358, and 392–394 are indicated). The aa sites bound by SARS-CoV and SARS-CoV-2 Spike [11] are indicated by red arrows. *SARS-CoV-specific interactions. (E) ACE2 sequences were cloned into the pDISPLAY expression construct in frame with an N-terminal signal peptide (the murine Ig κ-chain leader sequence) and HA-tag. (F) Expression of individual mammal or bird ACE2 proteins was confirmed at a whole cell level by western blot. (G) Flow cytometry was performed to examine surface expression of each ACE2 protein on non-permeabilised cells. For gated cells, the percentage positivity and MFI are plotted. The data underlying this figure may be found in S1 Data and S1 Raw Images. aa, amino acid; ACE2, angiotensin-converting enzyme 2; MFI, mean fluorescence intensity; RBD, receptor binding domain; SARS-CoV, SARS Coronavirus; SARS-CoV-2, SARS Coronavirus 2.

## Receptor screening using surrogate entry assays identifies SARS-CoV-2 Spike as a pan-tropic viral attachment protein

To examine the capacity of SARS-CoV-2 to enter cells bearing different ACE2 proteins, we used 2 related approaches. The first, based on the widely employed pseudotyping of lentiviral particles with SARS-CoV-2 Spike [9], mimics particle entry. The second approach, based on a quantitative cell–cell fusion assay, we routinely employ for the morbilliviruses [15], assessing the capacity of Spike to induce cell–cell fusion following receptor engagement. In both assays, we used a codon-optimised SARS-CoV-2 Spike expression construct as the fusogen, demonstrating robust and sensitive detection of either entry or fusion above background (S2A and S2B Fig). Supportive of our technical approach, replacing the human ACE2 signal peptide with that found in pDISPLAY had no effect on pseudotype entry or cell–cell fusion, nor did the addition of the HA-tag (S2A and S2B Fig). In addition, SARS-CoV-2 entry was shown only with human ACE2, but not with aminopeptidase N (APN) or dipeptidyl peptidase 4 (DPP4), the β-coronavirus group I and MERS-CoV receptors, respectively (S2 Fig), indicating high specificity in both assays. Using the classical pseudotype approach, which models particle engagement with receptors on the surface of target cells, we demonstrated that SARS-CoV-2 Spike has a relatively broad tropism for mammalian ACE2 receptors. Indeed, we observed that pangolin, dog, cat, horse, sheep, and water buffalo all sustained higher levels of entry than was seen with an equivalent human ACE2 construct (Fig 2A; left heatmap, first column). In contrast, all 3 bat ACE2 proteins we analysed (fruit bat, little brown bat, and horseshoe bat) sustained lower levels of fusion than was seen with human ACE2, as did turkey and chicken ACE2, the only nonmammalian proteins tested. In accordance with previously published data on SARS-CoV and SARS-CoV-2 usages of rodent ACE2 [1,16], rat ACE2 did not efficiently support SARS-CoV-2 particle entry. However, we observed that the ACE2 from hamsters did support pseudoparticle entry, albeit less efficiently than human ACE2.

In the separate cell–cell fusion assay, which provides both luminescence and fluorescence-based monitoring of syncytium formation, a similar trend was observed with expression of chinchilla, cat, pig, sheep, goat, water buffalo, and cattle ACE2 proteins on target cells all yielding higher signals than target cells expressing human ACE2 (Fig 2A; left heatmap, second column). Similar to the pseudotype assay, the expression of all 3 bat ACE2 proteins resulted in less cell–cell fusion than that seen with human ACE2. The heatmaps presented in Fig 2A represent the averaged results from 3 entirely independent pseudotype and cell–cell assay receptor usage screens (with representative data sets shown in S3 Fig).

Combining the results from all 6 screens demonstrates a significant degree of concordance between the 2 experimental approaches (Fig 2B). Although the high correlation (Pearson $r = 0.84$) was unsurprising, given that both approaches rely on the same Spike–ACE2 engagement, fusogen activation and membrane fusion process (albeit at virus–cell or cell–cell interfaces), there were some marked differences in sensitivity. For the pseudotype system, there was little appreciable evidence for particle entry above background levels with ferret, rat, chicken, turkey, or horseshoe bat ACE2, compared with either vector control (pDISPLAY) transfected cells (Fig 2A; bottom row) or ACE2-transfected cells infected with a "no glycoprotein"

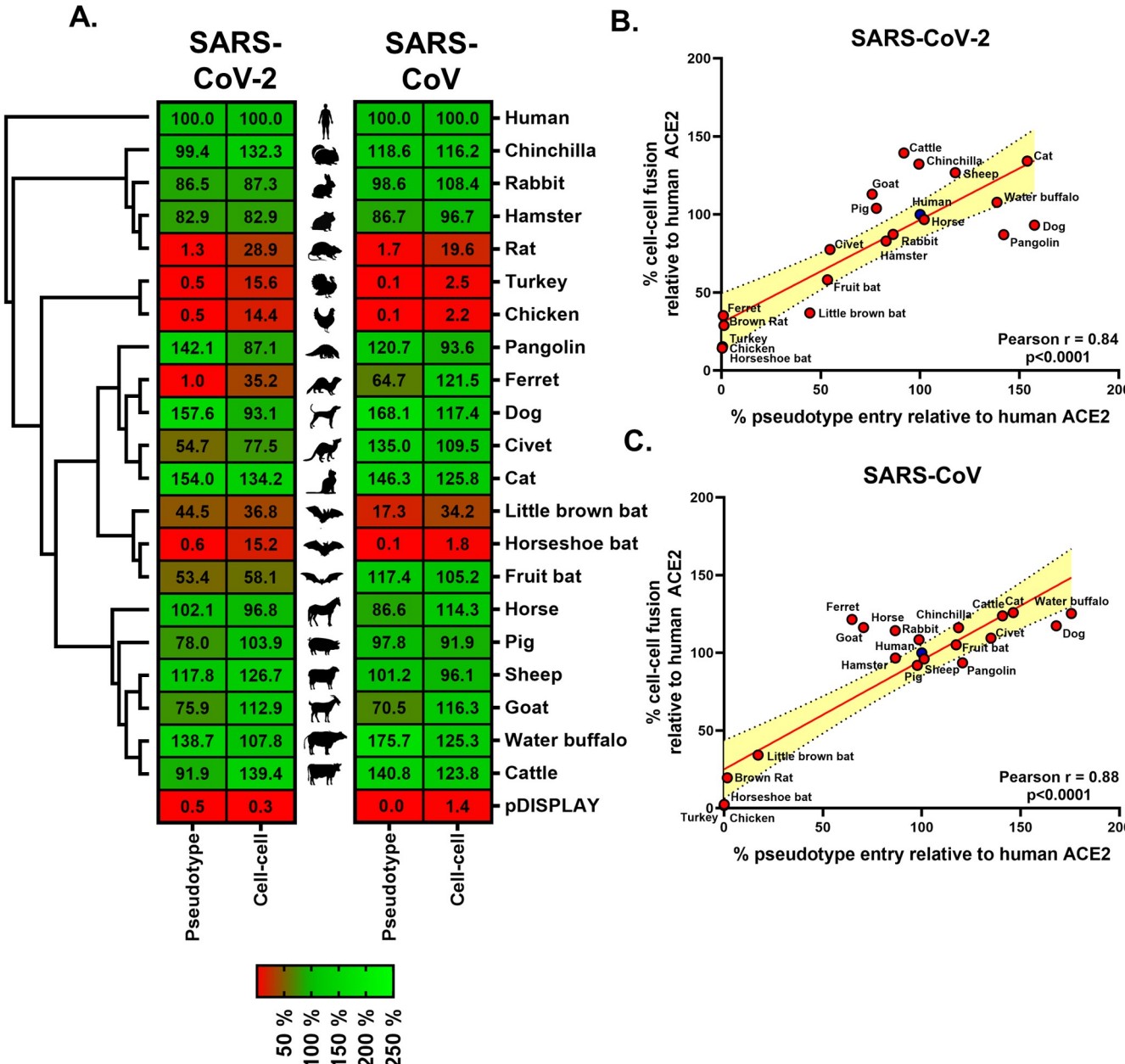

**Fig 2. Receptor screening using surrogate entry assays identifies SARS-CoV-2 Spike as a pan-tropic viral attachment protein.** (A) A heatmap illustrating the receptor usage profile of SARS-CoV-2 and SARS-CoV in pseudotype entry and cell–cell fusion assays with various mammalian and bird ACE2s. The data in each row are normalised to the signal seen for human ACE2 (top), with results representing the mean percentage calculated from 3 separate experiments performed on different days. A vector-only control (pDISPLAY) was added to demonstrate specificity. Mammalian and bird ACE2s are organised, top to bottom, based on their phylogenetic relationship (rectangular cladogram, left). The inter-experimental standard error of the mean for the pseudotype and cell–cell fusion assays ranged from 0.01% to 47.92% (median 10.73%) and 0.12% to 32.97% (median 5.43%), respectively. (B and C) For both SARS-CoV-2 and SARS-CoV, the respective cell–cell and pseudotype assay percentages for each ACE2 protein (relative to human ACE2) were plotted on an XY scatter graph, the Pearson correlation calculated and a linear line of regression fitted together with 95% confidence intervals. The data underlying this figure may be found in S1 Data. ACE2, angiotensin-converting enzyme 2; SARS-CoV, SARS Coronavirus; SARS-CoV-2, SARS Coronavirus 2.

pseudoparticle control, non-enveloped (NE) (S3 Fig). However, in the cell–cell system, all of these receptors permitted Spike-mediated fusion, above the background levels seen in pDIS-PLAY transfected cells (Fig 2A) or in effector cells not expressing SARS-CoV-2 Spike (S3 Fig;

no spike), albeit at levels significantly lower than that seen for human ACE2. This suggests that these receptors, whose structures are clearly not optimal for SARS-CoV-2 entry, are still bound by the Spike protein. Of note, the entry of SARS-CoV-2 and SARS-CoV is facilitated by the cellular serine protease transmembrane protease serine 2 (TMPRSS2), which primes the coronavirus Spike through specific cleavage events [8]. During the optimisation of our ACE2 receptor screening experiments, we also examined how transient expression of TMPRSS2 affects SARS-CoV-2 particle entry and cell–cell fusion. In our pseudoparticle entry experiments, overexpression of TMPRSS2 had a negligible impact on entry when cognate ACE2 receptors were expressed on target cells (S4A Fig; human, dog, cat, pig, and goat). We believe this is likely due to the saturation of pseudoparticle viral entry, a result of cognate ACE2 overexpression. Importantly however, receptors, which showed little evidence of supporting particle entry when expressed on their own, supported much more robust levels of entry when TMPRSS2 was co-expressed (S4A Fig; turkey and chicken), despite TMPRSS2 protein alone not supporting entry (S4B Fig). A similar trend was observed in our cell–cell fusion assay. Co-expression of human ACE2 and TMPRSS2 in target cells or separately trypsin treatment of Spike expressing effector cells both led to larger syncytia when compared with ACE2 alone, likely the result of direct activation of a greater percentage of the SARS-CoV-2 Spike found at the cell–cell interface (S4C Fig). However, TMPRSS2 co-expression significantly enhanced the use of non-cognate receptors (S4D Fig; fruit bat, ferret, turkey, and chicken), when compared with ACE2 or ACE2 and trypsin treatment. Since we were specifically interested in assessing nonhuman ACE2 interactions with SARS-CoV-2 Spike, we therefore did not include overexpressed TMPRSS2 in any of our host range receptor screening experiments (Fig 2).

To facilitate comparison with existing data for SARS-CoV, we also performed similar receptor screening experiments with SARS-CoV pseudotype and cell–cell assays (Fig 2A and 2C, right heatmap and S2 and S3 Figs). While the receptor usage profile of SARS-CoV correlates significantly with SARS-CoV-2, both in terms of pseudotype entry (S5A Fig; $r = 0.85$) and cell–cell fusion (S5B Fig; $r = 0.81$), there were interesting divergences. Although there was a similar restriction for bird and bat ACE2 proteins, our side-by-side comparison identified instances of varying restriction, with ferret, fruit bat, and civet ACE2 appearing to be preferentially used by SARS-CoV (Figs 2A and S5B). In summary, using 2 distinct technical approaches that monitor Spike-mediated receptor usage in a biologically relevant context, we provide evidence that SARS-CoV-2 has a broad tropism for mammalian ACE2s. These assays demonstrate correlation between ACE2 protein sequence and fusion by SARS-CoV or SARS-CoV-2 Spike protein, plus evidence of a low affinity of SARS-CoV Spike proteins for bird or rat ACE2 and varying levels of bat ACE2 utilisation.

## A cognate ACE2 receptor is required for SARS-CoV-2 infection

High-throughput and robust surrogate assays for SARS-CoV-2 viral entry only serve to model this process and can never completely replace live virus experiments. To this end, and in order to examine the permissiveness of nonhuman cell lines in our cell culture collection to SARS-CoV-2, we experimentally infected a range of animal cells including those established from birds, canids, rodents, ruminants, and primates (see S1 Table for details on species and cell type) with SARS-CoV-2 isolated from a patient in the United Kingdom (SARS-CoV-2 England-2/2020). Infection at a low multiplicity of infection (MOI) (0.001) failed to generate infectious virus in any of the cells tested, apart from 2 monkey cell lines (Vero E6 and Marc 145), in line with primate cells being used widely to propagate SARS-CoV-2 [17] (Fig 3A). Repeat infections at a higher MOI (1) in a subset of these cells (PK15, RK13, DF-1, and BHK-21) established evidence for a very low level of virus production only in the porcine cell line

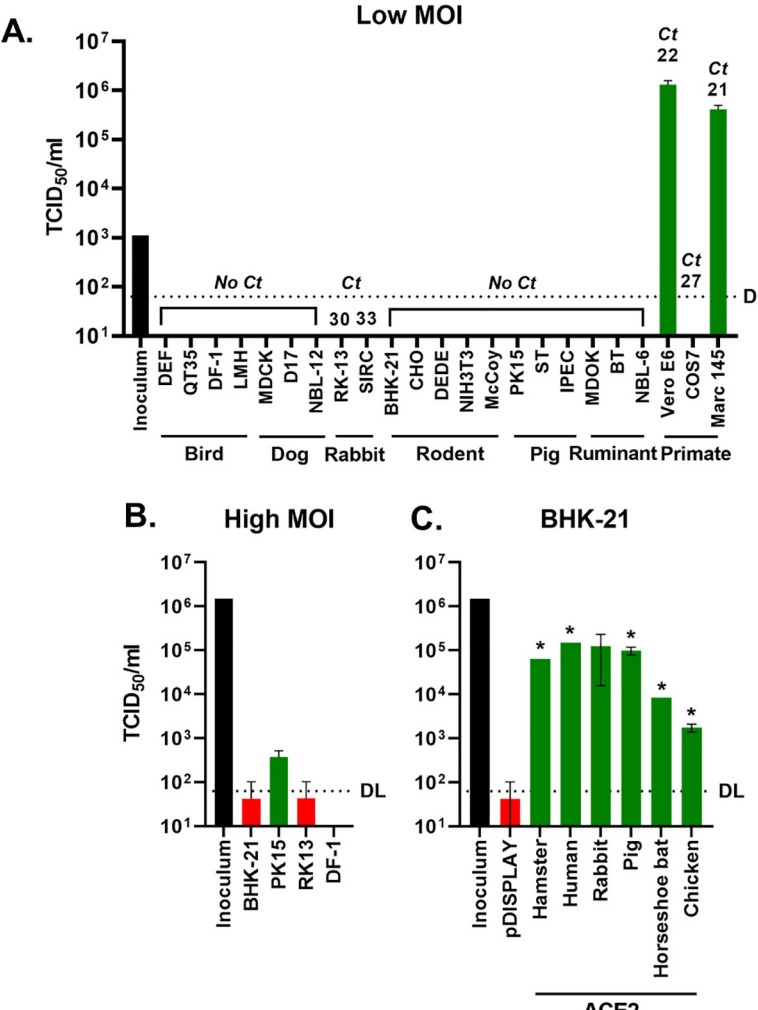

**Fig 3. A cognate ACE2 receptor is required for SARS-CoV-2 infection.** (A) Various cell lines derived from birds, dogs, rabbits, rodents, pigs, ruminants, and primates were experimentally infected with SARS-CoV-2 at a MOI of 0.001. At 72 h postinfection, the supernatants from cells were harvested and titred by TCID-50. For each cell line, RNA from uninfected cells was also extracted, and RT-qPCR was performed to detect ACE2 mRNA, with the value above each line indicating the cycle when PCR positivity was achieved (Ct). (B) Four of the same cell lines were infected again, this time at high MOI (1). (C) BHK-21 hamster cells were transiently transfected with ACE2 expression constructs (or a vector control [pDISPLAY]) before being infected with SARS-CoV-2 at high MOI (1). $^*p < 0.05$ Student $t$ test, compared with pDISPLAY. For all high MOI experiments, supernatant samples were harvested at 48 hpi for titration by TCID-50. The DL for the TCID-50 is indicated. In all experiments, the initial inoculum used for infection was titred, and infections were performed in duplicate, with error bars denoting standard deviation from the mean. The data underlying this figure may be found in S1 Data. ACE2, angiotensin-converting enzyme 2; BHK-21, baby hamster kidney 21; Ct, cycle threshold; DL, detection limit; MOI, multiplicity of infection; RT-qPCR, reverse transcription quantitative PCR; SARS-CoV-2, SARS Coronavirus 2; TCID50, 50% tissue culture infective dose.

PK15 (Fig 3B). Subsequent quantitative PCR (qPCR) analysis of ACE2 mRNA levels in the whole panel of cell lines, assayed using a novel panel of species-specific ACE2 primers, identified only 2 cell lines (Vero E6 and Marc 145) with cycle threshold (Ct) values less than 25 (Fig 3A), providing a strong correlative link between ACE2 receptor expression and successful virus infection.

We next sought to correlate the receptor usage results from our surrogate entry assays (Fig 2) with live virus infections. A baby hamster kidney cell line (BHK-21), which we established

as refractory to SARS-CoV-2 infection (Fig 3A and 3B), was transfected with vector alone (pDISPLAY) or a restricted panel of ACE2 constructs (hamster, human, horseshoe bat, rabbit, pig, and chicken) representing the spectrum of receptor usage (Fig 2A). Concurrent to the infections, the expression of ACE2 in equivalently transfected cells was confirmed by western blot, flow cytometry, and SARS-CoV-2 pseudotype infections (S6A–S6C Fig). Of note, for the live virus infections the high MOI (1), the inoculum was removed after 1 h with the cells thoroughly washed prior to incubation at 37°C. Accordingly, in the BHK-21 cells transfected with carrier plasmid, we saw very little evidence for virus infection and/or virus production, confirming that these cells do not natively support SARS-CoV-2 infection (Fig 3C). For the receptors where we had previously seen high levels of cell–cell fusion (hamster, pig, and rabbit), we observed robust viral replication (Fig 3C). Surprisingly, the 2 receptors included because of their "poor" usage by SARS-CoV-2 Spike (horseshoe bat and chicken ACE2, Fig 2A) were still able to support viral replication, albeit to a lower level (<5% of human ACE2 levels). Of note, regardless of the ACE2 species expressed, we saw very little evidence of cytopathic effect in the infected BHK-21 cells (S6D Fig), despite the release of infectious virus into the supernatant (S6E Fig). Lastly, focusing on the unexpected observation that chicken ACE2 permitted SARS-CoV-2 entry into cells, we investigated whether chicken embryonic fibroblast cells (DF-1) overexpressing chicken or human ACE2 could support viral replication. While western blot and flow cytometry demonstrated successful ACE2 overexpression (S7A and S7B Fig), we did not see any evidence of viral replication in these cells (S7C Fig). Since human ACE2 is efficiently used by SARS-CoV-2 Spike, we suspect that this is because of cell-specific deficiencies elsewhere in the virion entry pathway and/or a postentry block to viral replication in chicken cells. Interestingly, analysis of publicly available transcriptomic data from DF-1 cells [18] identified no detectable TMPRSS2 expression (S7D Fig). However, furin and cathepsin B mRNAs were both present. In summary, SARS-CoV-2 is able to use a range of nonhuman ACE2 receptors to enter cells. Furthermore, when a cognate ACE2 is provided, the virus can replicate efficiently in the normally refractory hamster cell line BHK-21.

## Amino acid substitutions within SARS-CoV-2 Spike RBD may have contributed to zoonotic emergence

The identification, isolation, and sequencing of SARS-CoV-2 progenitors in animal reservoirs or intermediate hosts could provide important information to explain how this virus emerged in human populations. However, as discussed previously, the most closely related virus strain currently available to researchers is RaTG13, isolated in *R. affinis* bats in 2013. A recent structure for RaTG13 Spike [12] allowed us to directly compare the RBDs of SARS-CoV-2 and RaTG13, identifying a high degree of structural conservation (Fig 4A). However, concurrent sequence analysis identified a number of variable residues within the RBD, which interact directly with ACE2 (Fig 4A, inset panel). Building on the hypothesis that the progenitor of SARS-CoV-2 was RaTG13-like, we next developed pseudotype and cell–cell fusion assays for RaTG13 Spike (S8A–S8C Fig) to examine the biological properties of this protein. Interestingly, RaTG13 Spike did not pseudotype as efficiently as SARS-CoV-2 or SARS-CoV, although it did retain significant fusogenicity in the cell–cell assay (Fig 4B). Screening with our panel of mammalian and bird ACE2 constructs also identified a phenotypically distinct pattern of receptor usage (Figs 4B and S8D and S8E) (Pearson *r* correlation RaTG13 versus SARS-CoV-2 = 0.573 for pseudotype entry and 0.510 for SARS-CoV, S8F and S8G Fig). Surprisingly, RaTG13 receptor usage of human ACE2 was still higher than that of horseshoe bat ACE2, although the human ACE2 values were significantly lower than those observed for SARS-CoV-2 and SARS-CoV (Fig 4B, blue data points). Importantly, the differing receptor usage

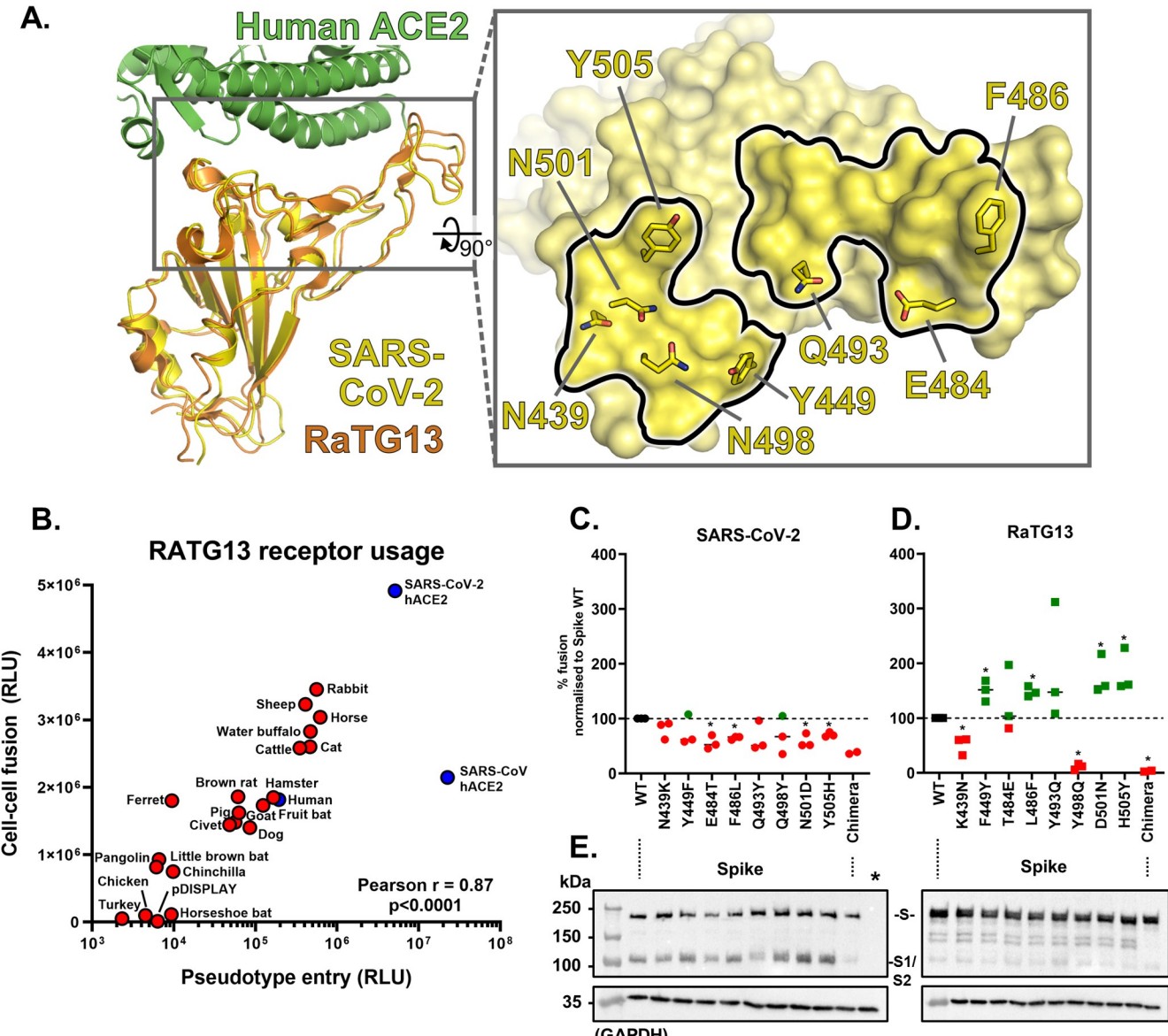

**Fig 4. Amino acid substitutions within SARS-CoV-2 Spike RBD may have contributed to zoonotic emergence.** (A) Comparison of the RBD structures of SARS-CoV-2 and RaTG13 Spike proteins [6,12] identified a high degree of structural similarity. Nevertheless, a number of amino acid changes between RaTG13 and SARS-CoV-2 were identified at residues interacting directly with ACE2 (according to [6]). SARS-CoV-2 N439, which does not interact directly with ACE2, was included because of its functional stabilisation role in the 498–505 loop, its previous identification within the SARS-CoV Spike RBD–ACE2 interface [19], and the N439K substitution present in RaTG13. (B) An XY scatter plot demonstrating the receptor usage profile of RaTG13 Spike in pseudotype (X) and cell–cell fusion assays (Y). Each point represents the mean signal seen from 3 experiments performed on separate days, with the human ACE2 highlighted in blue. Human ACE2 utilisation by SARS-CoV-2 and SARS-CoV Spike is also plotted for reference. In this graph, values are not normalised to human ACE2 since RaTG13 is, to our knowledge, not a human-tropic virus. (C and D) Specific amino acids within the RBDs of SARS-CoV-2 and RaTG13 Spike, which directly interact with ACE2 yet vary between these 2 sequences, were mutated to generate chimaeric SARS-CoV-2-RaTG13 Spike proteins. The cell–cell fusion activity of individual point mutants, as well as a full chimaera containing all 8 mutations (chimaera), was then examined using human ACE2 expressing target cells with activity normalised to the fusion seen with the wildtype (WT) viral glycoprotein of (C) SARS-CoV-2 or (D) RaTG13. The data points shown are the mean cell–cell fusion results seen in 3 completely independent experiments (green, increased relative activity; red, decreased relative activity). *$p < 0.05$ Student $t$ test, compared with WT. (E) Expression of the same mutants was analysed by western blot targeting the flag tag fused to these Spike proteins (S, full-length Spike; S1/S2, cleaved variant). Results are representative of protein expression experiments performed in duplicate, with a GAPDH loading control also shown. The data underlying this figure may be found in S1 Data and S1 Raw Images. ACE2, angiotensin-converting enzyme 2; GAPDH, glyceraldehyde 3-phosphate dehydrogenase; RBD, receptor binding domain; RLU, relative light units; SARS-CoV, SARS Coronavirus; SARS-CoV-2, SARS Coronavirus 2; WT, wildtype.

pattern of RaTG13 Spike, which contrasts with its high genetic similarity to SARS-CoV-2 Spike (98% at the amino acid level), indicated that amino acid variation within the RBD of this protein (Fig 4A) may play a deterministic role in host range. To identify which residues play a direct role in human ACE2 tropism, we generated a panel of RaT13 and SARS-CoV-2 Spike chimaeras, further examining our hypothesis that SARS-CoV-2 progenitors were RaTG13-like. Variable amino acids within the RBD were substituted with their corresponding residue in SARS-CoV-2 or RaTG13 Spike and the activity of these mutants assessed in a cell–cell fusion assay with human ACE2 (Fig 4C and 4D). Substitution of SARS-CoV-2 RBD residues proximal to the ACE2-binding surface [19] with those found in RaTG13 Spike was almost universally detrimental to human ACE2 receptor usage (Fig 4C), while an opposite trend was seen for RaTG13 (Fig 4D), aside from the substitutions at Spike positions 439 and 498, which were inhibitory in both contexts. In particular, the F449Y, L486F, Y493Q, D501N, and H505Y mutations to RaTG13 Spike markedly increased human ACE2 receptor usage. In contrast, the complete SARS-CoV-2/RaTG13 chimaeras containing all substitutions within ACE2-binding region of the RBD (Fig 4C and 4D; chimaera) were nonfunctional and had altered electrophoretic mobility in western blots (Fig 4E) suggesting that the deleterious mutations (K439N or Y498Q) have a dominant effect in inhibiting ACE2 utilisation, perhaps by impeding correct folding of the mutated RaTG13 and SARS-CoV-2 proteins, as well as potentially the cleavage of SARS-CoV-2 Spike into S1 and S2 (Fig 4E; left panel). If our hypothesis that SARS-CoV-2 arose from a RaTG13-like progenitor is correct, then mutations at these residues may have played an important role in the emergence of SARS-CoV-2 in the human population. Indeed, a similar pattern of mutations was identified for SARS-CoV during its emergence from its intermediate civet host [13,20], highlighting a potentially conserved mechanism for β-coronavirus adaptation to the human ACE2 receptor. Of note, the described mutants were constructed only in RaTG13 Spike expression plasmids, and assays were only performed in surrogate receptor usage assays, an approach we have previously used to safely interrogate the zoonotic potential of other viruses [15].

## Discussion

Recognising animals at risk of infection and/or identifying the original or intermediate hosts responsible for the SARS-CoV-2 pandemic are important goals for ongoing COVID-19 research. In addition, there is a requirement to develop appropriate animal models for infection that, if possible, recapitulate the hallmarks of disease seen in people. Importantly, high-resolution structures of human ACE2 in complex with the Spike RBD [5,6,10,11] can help us to understand the genetic determinants of SARS-CoV-2 host range and pathogenesis. In particular, differences in receptor usage between closely related host species or viruses provides an opportunity to pinpoint amino acid substitutions at the interaction interface that inhibit Spike protein binding and fusion, and ultimately determine host range.

One example of closely related ACE2 sequences differing in their utilisation by SARS-CoV-2 Spike comes from the comparison of rat and hamster ACE2. Although a number of animal models have been investigated for SARS-CoV-2, including nonhuman primates, ferrets, and cats [21,22], the use of small animals, in particular rodents, has proved more challenging as murine and rat ACE2 support lower levels of β-coronavirus entry [1,16]. For SARS-CoV, this problem was circumvented with the development of transgenic mice expressing human ACE2 [23] or mouse-adapted SARS-CoV [24,25], research that has recently been extended to SARS-CoV-2 [26]. Consistent with previously published data on SARS-CoV rodent ACE2 interactions, we showed that rat ACE2 does not support SARS-CoV-2-mediated fusion (Fig 2A). The hamster cell lines we used in our study (BHK-21 and Chinese hamster ovary (CHO)) are likely

refractory to infection simply because they express low levels of ACE2 mRNA (Fig 3A, qPCR data). However, our finding that hamster ACE2 allows the entry of SARS-CoV-2 (Fig 2A) indicates this animal is a suitable model for infection, consistent with recent in vivo studies demonstrating experimental infection of these animals [27]. Comparison of the hamster and rat sequences (Fig 5A) identified multiple substitutions at the RBD interaction interface that might explain this variable receptor tropism (listed as hamster to rat): Q24K, T27S, D30N, L79I, Y83F, and K353H. Except for L79I, which is similarly substituted in pangolin and pig ACE2, all of these substitutions are likely to reduce Spike RBD binding. Q24K and Y83F substitutions would both result in the loss of hydrogen bonds with the side chain of SARS-CoV-2 RBD residue N487 (Fig 5B). The T27S substitution would remove the threonine side chain methyl group that sits in a hydrophobic pocket formed by the side chains of RBD residues F456, Y473, A475, and Y489, and substitution of residue D30 (which is acidic in all ACE2 proteins efficiently utilised by SARS-CoV-2 Spike) to asparagine would remove the salt bridge formed with K417 of the RBD (Fig 5B). Furthermore, the K353H substitution would remove hydrogen bond interactions with the side chain of RBD Q498 and the backbone carbonyl oxygen of RBD G496, and neither of these could be formed by the shorter histidine side chain (Fig 5B). This K353H substitution is particularly noteworthy because mouse ACE2, which is also unable to efficiently bind SARS-CoV-2 Spike [1], also has a histidine at residue 353. However, introduction of the substitutions Q498Y and P499T in the Spike protein is sufficient to confer upon SARS-CoV-2 the ability to replicate in mice [26]. Introducing the larger tyrosine side chain at position 498 would likely restore a hydrogen bond with mouse ACE2 H353 and could facilitate hydrophobic interactions with the side chain of ACE2 Y41. Thus, multiple substitutions are predicted to inhibit Spike binding to rat ACE2 when compared with the closely related hamster protein, with K353H being of particular relevance (Fig 5B). Interestingly, along with *Mus musculus* [26], a similar ACE2 sequence-dependent susceptibility to SARS-CoV-2 has now been demonstrated for deer mice [28].

Another example of different receptor usage between closely related species can be seen with bat ACE2 (Figs 2A and 5A). The apparent lack of tropism for bat ACE2 proteins we observed was surprising as there is previous evidence of SARS-CoV-2 infection of bat ACE2 expressing cells in vitro [1], and in vitro binding experiments suggest that the SARS-CoV-2 RBD binds bat ACE2 with high affinity [29]. Since the exact origin of SARS-CoV-2 is currently unknown, but widely accepted to be a Chiroptera species, we included ACE2 proteins from a broad range of bats in our study. While none of these bat ACE2s supported SARS-CoV-2 fusion to the same levels as humans, there were dramatic differences in the ability of SARS-CoV-2 Spike to utilise the ACE2 from horseshoe bats versus fruit bats and little brown bats (Fig 2A). As discussed earlier, the closest known relative of SARS-CoV-2, RaTG13, was isolated from intermediate horseshoe bat (*R. affinis*). Unfortunately, the ACE2 sequence from this species was not available for use in our study; however, we did include an ACE2 from the closely related least horseshoe bat (*Rhinolophus pusillus*). Although this protein supported the lowest levels of fusion of any bat ACE2 tested in our study, it still supported a low level of SARS-CoV-2 replication with live virus (Fig 3C). As in rat ACE2, horseshoe bat and fruit bat ACE2 have a lysine residue at position 24 that would disrupt hydrogen bonding to N487 of the SARS-CoV-2 RBD and introduce a charge (Fig 5A and 5B). Little brown bats have the hydrophobic residue leucine at this position, which could not form the hydrogen bond to N487 but which is present in ACE2 from several species that support high levels of fusion, suggesting that the loss of the hydrogen bond is less deleterious to Spike protein binding than introduction of the lysine positive charge. Fruit bats conserve a T27, whereas little brown bats have the bulkier isoleucine residue and horseshoe bats have a bulky charged lysine residue in this position, both of which are likely to clash with the F456-Y473-A475-Y489 hydrophobic pocket of

## A.

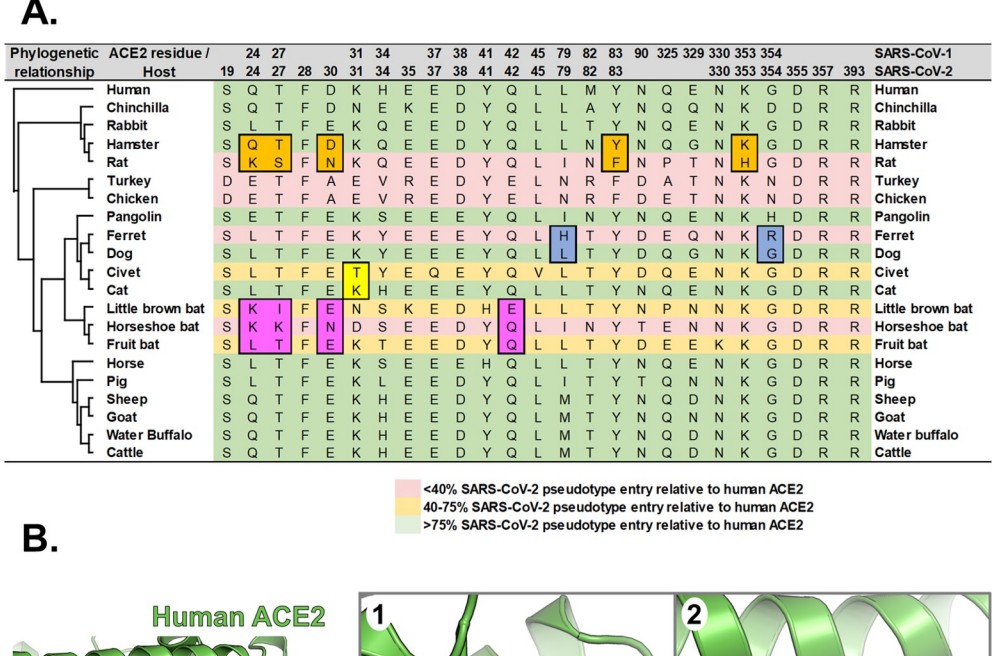

## B.

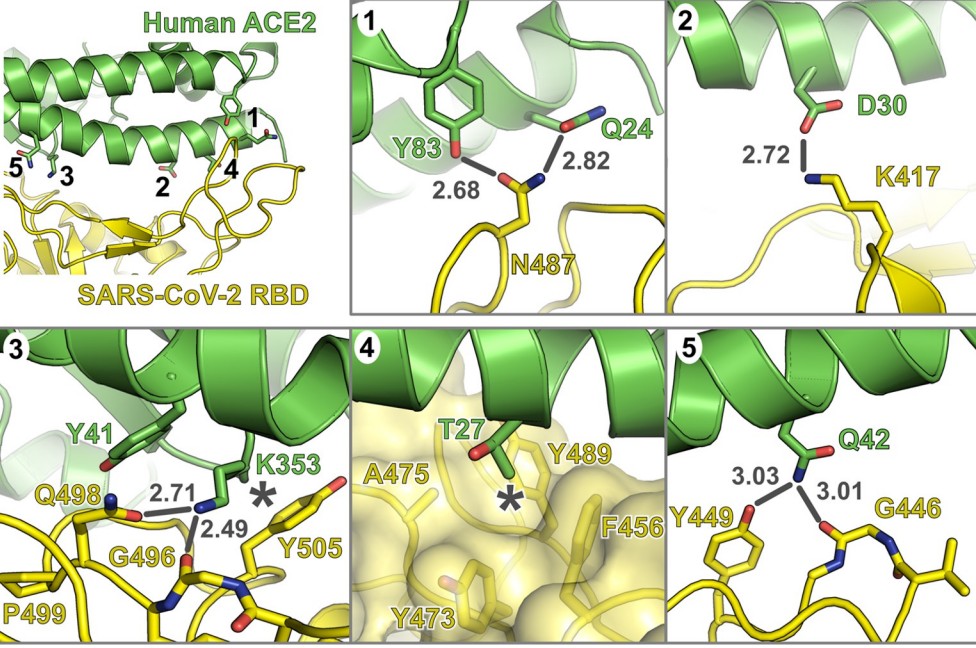

**Fig 5. Substitutions at the interface between SARS-CoV-2 RBD and mammalian ACE2 proteins impact receptor utilisation.** (A) Residues of mammalian ACE2 sequences used in this study that are predicted to interact with the RBD SARS-CoV and SARS-CoV-2, based on the structures of human ACE2 in complex with SARS-CoV [19] and SARS-CoV-2 [6]. Differences between closely related species that may impact RBD binding are highlighted. (B) Interface between human ACE2 (green) and SARS-CoV-2 RBD (yellow). Insets 1 to 5 show molecular interactions discussed in the main text. Bonds that may be disrupted are shown as grey lines, with bond distances in grey text, and hydrophobic interactions that may be disrupted are marked with asterisks. ACE2, angiotensin-converting enzyme 2; RBD, receptor binding domain; SARS-CoV, SARS Coronavirus; SARS-CoV-2, SARS Coronavirus 2.

the RBD, with the lysine substitution likely to be more deleterious due to the introduction of the positive charge. Like rats, horseshoe bat N30 would be unable to form a salt bridge with RBD K417. Substitution of Q42 with glutamate in little brown bat may be detrimental to Spike binding, as it would disrupt the hydrogen bond to the backbone carbonyl oxygen of RBD

residue G446. The other substitutions between bat ACE2 proteins and other mammals are likely to be benign. Little brown bats, horseshoe bats, pangolins, and horses all share a serine as ACE2 residue 34, suggesting that serine in this position does not abolish Spike binding, and it is likely that the threonine at this position (fruit bat ACE2) would likewise be tolerated. Similarly, the Y41H substitution present in little brown bat ACE2 is also present in horse ACE2, suggesting that it does not prevent binding. Therefore, all bat ACE2 proteins have substitutions that impair SARS-CoV-2 Spike binding to different degrees, but it seems likely that the E30N substitution (shared only by rat ACE2) or introduction of a charged lysine residue at position 27 (unique to horseshoe bats) are the most likely causes of the severely impaired binding of SARS-CoV-2 Spike to horseshoe bat ACE2.

Interestingly, a similarly "poor" tropism for bat ACE2 was also reported for SARS-CoV following its emergence in 2002 [30]. Specifically, coronaviruses closely related to SARS-CoV that were isolated directly from bats were shown to inefficiently use either human or civet ACE2 [30]. This is consistent with large shifts in receptor usage occurring during coronavirus species jumps, either directly into humans or more likely via intermediate reservoirs. During the SARS-CoV epidemic, where civets were identified as the intermediate reservoir of infection, a shifting pattern of increasing and decreasing ACE2 usage was observed in individual isolates of SARS-CoV taken from civets and humans (although they shared approximately 99% similarity to each other), providing evidence for adaptation to individual host receptors [13,20] with a particular focus on differential adaptation to human ACE2 residues K31 (T31 in civets) and K353. Interestingly, correlation analysis of SARS-CoV and SARS-CoV-2 pseudo-type entries highlighted civet ACE2 as being strongly favoured by SARS-CoV, perhaps a legacy of this period of adaptation in an intermediate host (S5A Fig). Although data analysis of this type between related viruses might represent a mechanism for identifying intermediate reservoirs, similar outliers that favoured SARS-CoV-2 entry were not evident in our study. As discussed above, the lack of similarly closely related SARS-CoV-2 isolates from this outbreak's origin in Hubei makes detailed interpretation of this virus's adaptation to human ACE2 difficult at this time. However, to address this gap in our understanding, we performed detailed analysis of the closest relative of SARS-CoV-2, RaTG13. Strikingly, RaTG13 was shown to have a distinct receptor usage pattern from SARS-CoV-2, with SARS-CoV-2 and SARS-CoV being significantly more related (S8F and S8G Fig; SARS-CoV-2 Pearson correlation $r = 0.57$ [RatG13] versus 0.83 [SARS-CoV]). This is despite SARS-CoV-2 and RaTG13 Spike being 98% identical at the amino acid level, with SARS-CoV-2 and SARS-CoV sharing only 77% identity. Our mutational analysis of the SARS-CoV-2 and RaTG13 RBD identified a number of residues that are important in conferring human ACE2 tropism to the Spike protein (Fig 4C and 4D). Structural studies of SARS-CoV-2 [6], and more recently RaTG13 [12], help to shed light on the interactions underpinning these phenotypic changes (Fig 6). An F449Y substitution in RaTG13 Spike would allow hydrogen bonds to be formed with the D38 and Q42 side chains of ACE2, with the loss of the tyrosine hydroxyl group explaining the small loss of function for the Y449F mutation in SARS-CoV-2 (Figs 4C and 4D and 6). The drop in SARS-CoV-2 fusion activity of the F486L mutant (Fig 4C) is likely due to reduced ACE2 affinity arising from weakened interactions between the shorter leucine side chain and a hydrophobic pocket on ACE2 formed by L79, M82, and Y83 side chains and the M82-Y83 peptide plane (Fig 6). The converse is seen with the RaG13 L486F mutant (Fig 4D), suggesting that an extended hydrophobic interaction at this locus is important in conferring tropism to the human ACE2 receptor. The reduced ACE2 utilisation by Q493Y-substituted SARS-CoV-2 Spike (Fig 4C) is likely due to the loss of a hydrogen bond with ACE2 E35 and steric clashes between the bulky tyrosine side chain and the side chain of ACE2 K31 (Fig 6), with the reciprocal change in RaTG13 (Y493Q) giving rise to increased fusion (Fig 4D). The Y505H substitution in

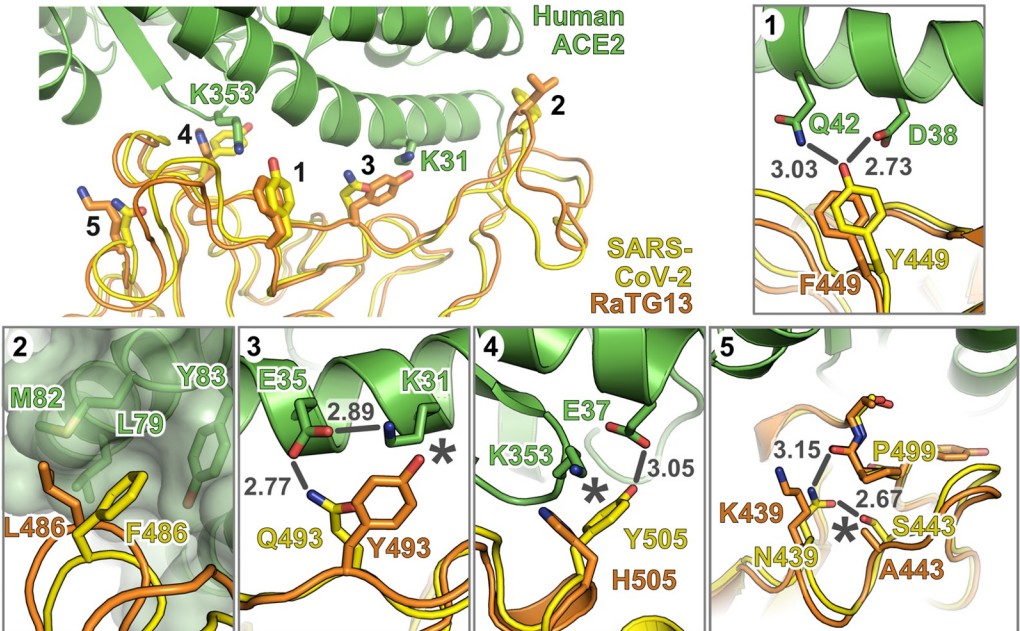

**Fig 6. Amino acid variation in SARS-CoV-2 and RaTG13 RBD impacts human ACE2 receptor utilisation.** Structure of RaTG13 RBD (orange) [12] superposed onto the structure of human ACE2 (green) in complex with the SARS-CoV-2 RBD (yellow) [10]. Selected RBD residues that promote association with ACE2 are highlighted, as are the ACE2 "hotspot" residues K31 and K353 [13]. Insets 1 to 5 show molecular interactions discussed in the main text. Bonds that may be disrupted are shown as grey lines, with bond distances in grey text, and hydrophobic interactions that may be disrupted or potential steric clashes are marked with asterisks. ACE2, angiotensin-converting enzyme 2; RBD, receptor binding domain; SARS-CoV-2, SARS Coronavirus 2.

SARS-CoV-2 Spike also reduced human ACE2 receptor usage, with a concordant increase in RaTG13 activity when the reciprocal H505Y substitution was introduced (Fig 4C and 4D). This can potentially be explained by the terminal hydroxyl group of Y505, which makes a hydrogen bond with ACE2 E37, and by the aromatic region of the tyrosine side chain that forms hydrophobic interactions with the alkyl region of the K353 side chain (Fig 6). The RaTG13 H505 side chain is too short to sustain this hydrogen bond and would make less extensive hydrophobic interactions; thus, the loss (or gain) of this hydrogen bond may decrease (or increase) human ACE2 binding. One site where reciprocal substitutions between SARS-CoV-2 and RaTG13 do not give rise to concomitant loss or gain of receptor utilisation is residue 439, as both the SARS-CoV-2 N439K and RaTG13 K439N mutants exhibit reduced fusion (Fig 4C and 4D). Residue 439 does not interact directly with ACE2, but in SARS-CoV-2, the N439 side chain forms hydrogen bonds with the side chain of SARS-CoV-2 S443 and with the backbone carbonyl oxygen of SARS-CoV-2 P499 (Fig 6). The N439K substitution may modestly alter the conformation and increase the flexibility of the 437–448 and 498–505 loops in SARS-CoV-2, both of which interact with ACE2, leading to modestly decreased receptor utilisation (Fig 4C). In RaTG13, residue 443 is an alanine and would thus be unable to form a hydrogen bond with the asparagine side chain introduced by the K439N substitution (Fig 6), potentially leading to a structural rearrangement of the RaTG13 437–448 loop that is deleterious to ACE2 binding (Fig 4D). It is notable that several of the SARS-CoV-2 residues that are important for human ACE2 receptor utilisation bind in close proximity to the 2 hotspots for human ACE2 adaptation (K31 and K353) that were identified previously for SARS-CoV (Fig 6, [13]; see also discussion above). Although researchers do not have access to SARS-CoV-2 viruses, or their progenitors, from wildlife or intermediate hosts, our data

highlight that β-coronavirus adaptation to human ACE2 might follow a conserved mechanism. The high correlation between SARS-CoV and SARS-CoV-2 host range, despite their genetic divergence, also indicates zoonotic β-coronaviruses may have a conserved and identifiable pattern of receptor usage. This technique may prove useful for determining the zoonotic potential of other emerging viruses in the future.

In the process of finalising this manuscript, related studies examining the receptor usage of various nonhuman ACE2s were published or released as preprints [31–33]. While these studies did not perform specific examination of cell–cell fusion, there is a strong correlation between their findings and ours, namely, the broad tropism of SARS-CoV-2 Spike, albeit with specific examples of restriction in rodent [33], bat [32], and bird [31] species. Our findings are supported by the results of experimental SARS-CoV-2 infections, which showed that cats [22,34], hamsters [27], and fruit bats [34] are susceptible to infection, while chickens are not [22]. Elsewhere, evidence for the broad host tropism of SARS-CoV-2 includes the successful establishment of infection models in rhesus macaques, cynomolgus macaques, and African green monkeys [35–37]. However, certain animals where we demonstrated efficient ACE2 receptor usage, e.g., pigs and dogs (Fig 2A), appear less susceptible to experimental challenge [22,34]. It should be noted that particle entry represents only the first step in zoonotic spillover with multiple virus–host interactions combining to define virus host range and pathogenesis. It may be that the intracellular environment of specific hosts, e.g., pigs, cannot sustain SARS-CoV-2 infection, either through the absence of an important virus–host interaction or the presence of effective mechanisms of innate immune restriction. In this case, SARS-CoV-2 may enter cells efficiently but fail to replicate to significant levels to support onward transmission, bring about clinical signs, or induce immunopathological sequelae. In our experiments, the chicken DF-1 cell line was refractory to infection, even when a cognate ACE2 (human) was overexpressed in these cells. This may be due to an absence of TMPRSS2 in DF-1s (S7D Fig), although it is currently unclear whether this protease is absolutely required for SARS-CoV-2 entry since alternative pathways for Spike processing have been identified [38]. The inefficient replication of SARS-CoV-2 in chickens and eggs [22,34,39] indicates that multiple restrictions to efficient viral replication may exist in chicken cells. That being said, for some animals such as mice [26], it is now clear that inefficient ACE2 receptor usage is the primary barrier to infection. In summary, although the potential for SARS-CoV-2 to spread by reverse zoonosis appears real, and there is evidence for community-based infections in mink, tigers, cats, and dogs [40–42], the epidemiological significance of these infections remains to be determined. More thorough investigation, including heightened virus surveillance and detailed experimental challenge studies, are required to ascertain whether livestock and companion animals could act as reservoirs for this virus. From a molecular perspective, differences in receptor usage between host and virus are likely the result of amino acid variation within the Spike RBD–ACE2 interface. Indeed, our mutational analysis of SARS-CoV-2 and RaTG13 Spike confirmed that changes within this region have a significant impact on human ACE2 interactions, changes which are implicated in the emergence of SARS-CoV-2 in the human population.

## Methods

### Cell lines

Cell lines representing a broad range of animal species were used to determine the host range/tropism of SARS-CoV-2 (S1 Table) (Cell Culture Central Services Unit, The Pirbright Institute, UK). Cells were maintained in complete medium supplemented with either 10% horse or bovine viral diarrhea virus (BVDV) and foot and mouth disease virus (FMDV)-negative foetal bovine serum (FBS) (Gibco, Thermo Fisher Scientific, USA), 1% nonessential amino acids, 1 mM sodium pyruvate solution (Sigma-Aldrich, Germany), 2 mM L-glutamine (Sigma-

Aldrich), and 1% penicillin/streptomycin, 10,000 U/mL (Life Technologies, Thermo Fisher Scientific, USA). Additional supplements and cell culture medium for each cell line are summarised in S1 Table. All cells were incubated at 37°C in a humidified atmosphere of 5% $CO_2$.

Cells used for entry studies or fusion assays: HEK293T cells stably expressing a split *Renilla* luciferase-green fluorescent protein (GFP) plasmid (rLuc-GFP 1–7) and BHK-21 cells were maintained in Dulbecco's Modified Eagle Medium (DMEM)-10%: DMEM (Sigma-Aldrich) supplemented with 10% FBS (Life Science Production, UK), 1% 100 mM sodium pyruvate (Sigma-Aldrich), 1% 200 mM L-glutamine (Sigma-Aldrich), and 1% penicillin/streptomycin, 10,000 U/mL (Life Technologies). Stable cell lines were generated, as described previously, using a lentiviral transduction system under 1 μg/mL puromycin (Gibco) selection [43,44].

### Viruses and virus titre quantification

SARS-CoV-2 England-2/2020 was isolated from a patient in the UK, and a passage 1 stock was grown and titred in Vero E6 cells by Public Health England (PHE) (kindly provided to The Pirbright Institute by Prof. Miles Carroll). A master stock of virus was passaged to P2 in Vero E6 at a MOI of 0.001 in DMEM/2% FBS and used for all virus assays, following a freeze–thaw cycle at −80°C. Stocks were titred by plaque assay on Vero E6 cells using a 1× minimum essential media (MEM)/0.8% Avicel/2% FBS overlay, fixed using formaldehyde and stained using 0.1% Toluidine blue. All infections were performed in the Advisory Committee on Dangerous Pathogens (ACDP) hazard group 3 (HG3) facilities by trained personnel.

### Plasmids

Codon-optimised ACE2-expressing plasmids from a range of animal species were synthesised and cloned into pDISPLAY (BioBasic, Canada) (S2 Table). Codon-optimised SARS-CoV, SARS-CoV-2, and RaTG13 Spike sequences were synthesised and cloned into pcDNA3.1+ (BioBasic) (S3 Table).

### Infections

Initial screen: Cells listed in S1 Table were seeded at a density of $1 \times 10^5$ cells/well in a 24-well plate (Nunc, Thermo Fisher Scientific, USA) and, 24 h later, infected with SARS-CoV-2 England-2/2020. Briefly, media was removed, and the cells washed once with complete DMEM supplemented with 2% FBS. Cells were then infected at MOI 0.001 and incubated at 37°C for 1 h. Following this, the inoculum was removed, and cells were washed twice with PBS, complemented with cell maintenance media and incubated for 72 h at 37°C. Supernatant was collected at 72 h postinfection and frozen at −80°C until required. Cells were fixed with formaldehyde for 30 min and then stained with 0.1% Toluidine blue (Sigma-Aldrich).

Receptor usage screen: BHK-21 and DF-1 cells were plated at $1 \times 10^5$ cells/well in 24-well plates (Nunc). The following day, cells were transfected with 500 ng of a subset of ACE2 expression constructs (human, hamster, rabbit, pig, chicken, and horseshoe bat) or mock transfected with an empty vector (pDISPLAY) in OptiMEM (Gibco) using TransIT-X2 transfection reagent (Mirus Bio, USA) according to the manufacturer's recommendations. Following this, cells were infected at MOI 1 as described above, and supernatants were collected at 72 h postinfection and frozen at −80°C until required.

### SARS-CoV-2 and SARS-CoV pseudoparticle infections

Pseudoparticle generation: Lentiviral-based pseudoparticles were generated in HEK293T producer cells and seeded in 6-well plates at $7.5 \times 10^5$/well 1 day prior to transfecting with the

following plasmids: 600 ng p8.91 (encoding for HIV-1 gag-pol), 600 ng CSFLW (lentivirus backbone expressing a firefly luciferase reporter gene), and 500 ng of SARS-CoV-2 Spike, SARS-CoV Spike or RaTG13 Spike in OptiMEM (Gibco) (S3 Table) with 10 μL polyethylenei-mine (PEI), 1 μg/mL (Sigma) transfection reagent. No glycoprotein controls (NE) were also set up using empty plasmid vectors (500 ng pcDNA3.1), and all transfected cells were incubated at 37˚C, 5% $CO_2$. The following day, the transfection mix was replaced with DMEM-10% and pooled harvests of supernatants containing SARS-CoV-2 pseudoparticles (SARS-CoV-2 pps), SARS-CoV pseudoparticles (SARS-CoV pps), and RaTG13 pseudoparticles (RaTG13 pps) were taken at 48 and 72 h post-transfection, centrifuged at $1,300 \times g$ for 10 min at 4˚C to remove cellular debris, aliquoted and stored at −80˚C. For the initial optimisation of pseudo-particle activity, 3 conditions were tested: (1) ACE2 expression only: HEK293T target cells transfected with 500 ng of a human ACE2 expression plasmid (Addgene, USA) and were seeded at $2 \times 10^4$ in 100 μL DMEM-10% in a white-bottomed 96-well plate (Corning, USA) 1 day prior to infection; (2) Spike activation with TMPRSS2: HEK293T target cells were trans-fected with 25 ng TMPRSS2 alongside human ACE2 as above; (3) Spike activation by trypsin treatment: viral pseudoparticles were treated with 2.5 mg/mL trypsin for 1 h at 37˚C before addition to target cells overexpressing human ACE2. SARS-CoV-2 pps, SARS-CoV pps, and RaTG13 pps, along with their respective NE controls were titrated 10-fold on target cells and incubated for 72 h at 37˚C, 5% $CO_2$. To quantify firefly luciferase, media was replaced with 50 μL Bright-Glo substrate (Promega, USA), diluted 1:2 with serum-free, phenol red-free DMEM, incubated in the dark for 2 min and read on a Glomax Multi+ Detection System (Promega).

Receptor usage screens: BHK-21 cells were seeded in 48-well plates at $5 \times 10^4$/well in DMEM-10% 1 day prior to transfection with 500 ng of different species, ACE2-expressing constructs or empty vector (pDISPLAY) (S2 Table) in OptiMEM and TransIT-X2 (Mirus Bio) transfection reagent according to the manufacturer's recommendations. The next day, cells were infected with SARS-CoV-2 pp/SARS-CoV pp equivalent to $10^6$ to $10^7$ relative light units (RLU), or their respective NE controls at the same dilution and incubated for 48 h at 37˚C, 5% $CO_2$. To quantify firefly luciferase, media was replaced with 100 μL Bright-Glo substrate (Pro-mega) and diluted 1:2 with serum-free, phenol red-free DMEM. Cells were resuspended in the substrate, and 50 μL was transferred to a white-bottomed plate in duplicate. The plate was incubated in the dark for 2 min and then read on a Glomax Multi+ Detection System (Pro-mega) as above. Comma-separated values (CSV) files were exported onto a universal serial bus (USB) flash drive for analysis. Biological replicates were performed 3 times.

## Cell–cell fusion assays

HEK293T rLuc-GFP 1–7 [45] effector cells were transfected in OptiMEM (Gibco) using Tran-sit-X2 transfection reagent (Mirus), as per the manufacturer's recommendations, with SARS-CoV-2 (1,000 ng), SARS-CoV (1,000 ng), RaTG13 (1,000 ng) (S3 Table), or SARS-CoV-2 and RaTG13 mutants (S4 Table) alongside mock-transfection with empty plasmid vector (pcDNA3.1+). BHK-21 target cells were co-transfected with 500 ng of different ACE2-expres-sing constructs (S2 Table) and 250 ng of rLuc-GFP 8–11 plasmid. In the initial fusion assay optimisation stage, protease activation was tested in 3 different conditions:(1) overexpression of ACE2-expresssing constructs only in target cells; (2) co-transfection of 25 ng of human TMPRSS2 in target cells; or (3) effector cells were washed twice with PBS and incubated with 3 μg/mL of Tosyl phenylalanyl chloromethyl ketone (TPCK)-treated trypsin (Sigma-Aldrich) for 30 min at 37˚C before resuspending in phenol red-free DMEM-10%. For the subsequent receptor usage screens, trypsin treatment was used for Spike activation. Target cells were

washed once with PBS and harvested with 2 mM EDTA in PBS before coculture with effector cells at a ratio of 1:1 in white 96-well plates to a final density of $4 \times 10^4$ cells/well in phenol red-free DMEM-10%. Quantification of cell–cell fusion was measured based on *Renilla* luciferase activity, 24 h later by adding 1 μM of Coelenterazine-H (Promega) at 1:400 dilution in PBS. The plate was incubated in the dark for 2 min and then read on a Glomax Multi+ Detection System (Promega) as above. CSV files were exported onto a USB flash drive for analysis. GFP fluorescence images were captured every 2 h for 24 h using an Incucyte S3 real-time imager (Essen Bioscience, Ann Arbor, Michigan, USA). Cells were maintained under cell culture conditions as described above. Assays were set up with 3 or more biological replicates for each condition, with each experiment performed 3 times.

## Western blotting

BHK-21 cells were transfected using Transit-X2 transfection reagent (Mirus Bio), as per the manufacturer's instructions with 500 ng of different ACE2-expressing constructs (S2 Table) or mock-transfected with empty plasmid vector (pDISPLAY). HEK293T cells stably expressing rLuc-GFP 1–7 were transfected with wild-type SARS-CoV-2 or RaTG13 Spike plasmids or mutants with amino acid substitutions (S4 Table). All protein samples were generated using 2× Laemmli buffer (Bio-Rad, USA) and reduced at 95˚C for 5 min 48 h post-transfection. Samples were resolved on 7.5% acrylamide gels by SDS-PAGE, using semidry transfer onto nitrocellulose membrane. Blots were probed with mouse anti-human influenza hemagglutinin tag (HA) primary antibody (Miltenyi Biotech, Germany) for ACE2-transfected cells at 1:1,000 or with rat anti-FLAG primary antibody (BioLegend, USA) for Spike transfected cells at 1:1,000 in PBS-Tween 20 (PBS-T, 0.1%) with 5% (w/v) milk powder overnight at 4˚C. Blots were washed in PBS-T and incubated with anti-mouse secondary antibody conjugated with horseradish peroxidase (Cell Signalling, USA) at 1:1,000 in PBS-T for 1 h at room temperature. Membranes were exposed to Clarity Western ECL substrate (Bio-Rad Laboratories) according to the manufacturer's guidelines and exposed to autoradiographic film.

## Flow cytometry

BHK-21 cells were transfected using Transit-X2 transfection reagent (Mirus Bio), as per the manufacturer's instructions with 500 ng of each ACE2-expressing construct (S2 Table) or mock-transfected with empty plasmid vector (pDISPLAY) for 48 h. Cells were resuspended in cold PBS and washed in cold stain buffer (PBS with 1% BSA (Sigma-Aldrich), 0.01% $NaN_3$ and protease inhibitors (Thermo Fisher Scientific, USA)). Cells were stained with anti-HA Phycoerythrin (PE)-conjugated antibody (Miltenyi Biotech) at 1:50 dilution for $1 \times 10^6$ cells for 30 min on ice, washed twice with stain buffer, and fixed in 2% paraformaldehyde for 20 min on ice. Fixed cells were resuspended in PBS before being analysed using the MACSQuant Analyzer 10 (Miltenyi Biotech), and the percentage of PE-positive cells was calculated by comparison with unstained and stained mock-transfected samples. Positive cells were gated as represented in S1 Fig, and the same gating strategy was applied in all experiments.

## RNA extraction and ACE2 qPCR quantification

Total cellular RNA was extracted from cell lines in S1 Table using a QIAGEN RNeasy RNA extraction kit (Qiagen, Germany), and mRNA was then detected with SYBR-Green–based qPCR (New England Biolabs, NEB, USA), using a standard curve for quantification on a QuantStudio 3 (Thermo Fisher Scientific, USA) thermocycler. Luna Universal qPCR Master Mix (NEB) was used to quantify mRNA levels for each cell line. RNA was first transcribed using SuperScript II Reverse Transcriptase (Thermo Fisher Scientific), with oligo dT primers

and 50 ng of input RNA in each reaction. All the reactions were carried out following the manufacturer's instructions and in technical duplicate, with the melt curves analysed for quality control purposes. Conserved cross-species ACE2 primers used for each cell line are found in S5 Table.

### Structural analysis

Molecular images were generated with an open-source build of PyMOL (Schrödinger, USA) using the crystal structure of SARS-CoV-2 RBD in complex with human ACE2 (protein data bank (PDB) ID 6M0J) [10] that had been further refined by Dr Tristan Croll, University of Cambridge, (https://twitter.com/CrollTristan/status/1240617555510919168) or the cryo-electron microscopy (EM) structure of RaTG13 Spike [12] superposed onto the SARS-CoV-2 RBD plus ACE2 complex using the secondary structure mapping (SSM) tool in the Crystallographic Object-Oriented Toolkit (COOT) software [46]. To analyse the conservation of mammalian ACE2 receptor sequences, representative sequences were identified via a position-specific iterative (PSI)–BLAST [47] search of the UniRef90 database [48] and filtering for the class Mammalia (taxid: 40674). The selected sequences were aligned using Multiple Alignment using Fast Fourier Transform (MAFFT) [49], and evolutionary conservation of amino acids was mapped onto the ACE2 structure using ConSurf software [50], implementing a Bayesian framework to estimate the evolutionary rate of each amino acid in the sequence where slowly evolving residues are conserved and fast-evolving residues are divergent.

### Phylogenetic analysis

ACE2 amino acid sequences were translated from predicted mRNA sequences or protein sequences (S2 Table). The predicted guinea pig mRNA sequence was more divergent than expected and contained a premature stop codon. For the purposes of this research, 5 single nucleotides were added, based on the most closely related sequence (chinchilla), to allow a full-length mature protein to be synthesised. It is not clear if the guinea pig has a functional ACE2, or if the quality of the genomic data is very low, but overall confidence in this sequence is low. The other divergent sequence was turkey as the 3′ end was not homologous with other vertebrate ACE2 receptors. This appeared to be a mis-annotation in the genome as the 3′ end showed very high identity to the collectrin gene. The missing 3′ of the gene was found in the raw genome data and assembled with the 5′ region to make a full ACE2 sequence. Twenty-three nucleotide base pairs were missing between these regions; these were taken from chicken as the most closely related sequence. Phylogenetic analysis of the final dataset was inferred using the neighbor-joining method [51] conducted in MEGA7 [52] with ambiguous positions removed. The tree is drawn to scale, and support was provided with 500 bootstraps.

### Data handling and statistical analysis

GraphPad Prism v8.2.1 (GraphPad Software, USA) was used for graphical and statistical analysis of data sets. Flow cytometry data was analysed using FlowJo software v10.6.2 (Becton, Dickinson & Company, USA).

## Supporting information

**S1 Fig. Gating strategy for flow cytometry analysis of ACE2-expressing constructs.** BHK-21 cells were transfected with a panel of species-specific ACE2-expressing constructs (see S2 Table). Cells were surface-stained with anti-HA PE-conjugated antibody. Live and singlet BHK-21 were gated as PE-positive, relative to mock-transfected cells (top panel).

Representative datasets are shown for human, goat, and guinea pig ACE2 surface staining (bottom panel). ACE2, angiotensin-converting enzyme 2; BHK-21, baby hamster kidney; FSC-A, forward scatter area; HA, human influenza hemagglutinin tag; PE, Phycoerythrin. (TIF)

**S2 Fig. Establishment of SARS-CoV-2 and SARS-CoV entry assays.** (A–D) Pseudotype and cell–cell fusion assays were established for SARS-CoV-2 (A and B) and SARS-CoV (C and D) using multiple internal controls. For the pseudotype assays NE lentiviral particles were generated, i.e., vector plasmid in place of a viral glycoprotein, to examine background levels of pseudoparticle entry. For the cell–cell fusion assay, mock-transfected effector cells were used (No Spike) to examine background levels of cell–cell fusion. In all subsequent experiments, "NE" and "No Spike" controls were compared against SARS-CoV-2 pseudoparticles or SARS-CoV-2 Spike-expressing effector cells (see S3 Fig). To validate our pDISPLAY approach, cells were transfected with expression constructs for full-length untagged hACE2 (FL) or a human ACE2 where the signal peptide was replaced with the murine Ig κ-chain leader sequence (hACE2) and the protein was tagged with the HA-epitope tag. In both instances, the corresponding vector controls, pcDNA3.1 and pDISPLAY, were separately transfected for comparison. The specificity of the SARS-CoV-2 and SARS-CoV assays were further confirmed by comparing hACE2-mediated fusion to human aminopeptidase N (hAPN) or dipeptidyl peptidase 4 (hDPP4) fusion, the coronavirus group I and MERS-CoV receptors, respectively. Lastly, in all assays, target cells representing un-transfected cells (mock) were also included. For pseudotype and cell–cell fusion assays, luciferase assays were performed in duplicate and triplicate, respectively, with the error bars denoting standard deviation. The data underlying this figure may be found in S1 Data. ACE2, angiotensin-converting enzyme 2; HA, human influenza hemagglutinin tag; hACE2, human ACE2; hAPN, human aminopeptidase N; hDPP4, human dipeptidyl peptidase 4; NE, non-enveloped; SARS-CoV, SARS Coronavirus; SARS-CoV-2, SARS Coronavirus 2. (TIF)

**S3 Fig. SARS-CoV-2 and SARS-CoV receptor usage screening.** As per S2 Fig NE and No Spike controls were included in all assays, as well as a vector-only control (pDISPLAY). For pseudotype and cell–cell fusion assays, luciferase assays were performed in duplicate and triplicate, respectively, with the error bars denoting standard deviation. Representative data sets from individual experiments are shown; however, the heatmaps and XY correlative plots in Figs 2 and S5 summarise the results from 3 independent experiments performed on separate days. The data underlying this figure may be found in S1 Data. ACE2, angiotensin-converting enzyme 2; NE, non-enveloped; RLU, relative light units; SARS-CoV, SARS Coronavirus; SARS-CoV-2, SARS Coronavirus 2. (TIF)

**S4 Fig. TMPRSS2 protease overexpression masks Spike-ACE2 receptor usage restrictions.** (A) Overexpression of human TMPRSS2 in target cells expressing various host ACE2 proteins has a negligible impact on SARS-CoV-2 pseudotype entry when a cognate ACE2 is present (human, dog, cat, pig, or goat); however, it disproportionately enhances entry with non-cognate ACE proteins (turkey and chicken), compared with ACE2 expression alone. Experiments were performed in biological triplicate and the mean RLU plotted, together with error bars denoting standard deviation. (B) Expression of TMPRSS2 alone does not support SARS-CoV-2 entry pseudotypes. NE HIV1 pseudotypes. (C) Trypsin treatment of effector cells or TMPRSS2 overexpression in the context of ACE2 leads to larger syncytia in SARS-CoV-2 cell–cell fusion assays. Trypsin treatment also increases SARS-CoV syncytium size, but TMPRSS2

expression observably reduces syncytia number. (D) Cell–cell fusion assay signals are higher when TMPRSS2 is co-expressed in target cells, including in target cells expressing non-cognate ACE2 proteins, e.g., turkey and chicken. Results represent the raw RLU signals from 3 independent experiments with the mean signal plotted and error bars denoting standard deviation. The data underlying this figure may be found in S1 Data. ACE2, angiotensin-converting enzyme 2; RLU, relative light units; SARS-CoV, SARS Coronavirus; SARS-CoV-2, SARS Coronavirus 2; TMPRSS2, transmembrane protease serine 2.
(TIF)

**S5 Fig. Correlating SARS-CoV-2 and SARS-CoV pseudotype and cell–cell fusion receptor usage.** The receptor usage data for SARS-CoV-2 and SARS-CoV was examined by separately comparing the pseudotype (A) or cell–cell fusion (B) assay results on XY scatter plots. Each point represents the mean activity calculated from 3 independent experiments. The Pearson correlation was calculated, and a linear line of regression fitted together with 95% confidence intervals. All values are plotted relative to the entry or cell–cell fusion recorded for human ACE2 (blue circles). The data underlying this Figure may be found in S1 Data. ACE2, angiotensin-converting enzyme 2; SARS-CoV, SARS Coronavirus; SARS-CoV-2, SARS Coronavirus 2.
(TIF)

**S6 Fig. Experimental infection of cell lines overexpressing vertebrate ACE2 proteins.** (A) SARS-CoV-2 pseudotype entry was assayed in BHK-21 transfected cells overexpressing ACE2 from the indicated species. Pseudotype infections were performed in triplicate, and the mean value was plotted on a heatmap following normalisation to human ACE2. Similarly, transfected target cells were lysed, and the ACE2 expression was analysed by western blot (B) or flow cytometry (C). (D) In parallel, BHK-21 cells were transfected with various ACE2-expressing constructs and infected with SARS-CoV-2 at an MOI of 1. Cells were fixed and stained at 48 hpi. (E) Prior to fixation, the supernatants from infected BHK-21 cells were removed for quantification of released virus by TCID-50. Representative images of these titrations, performed on Vero E6 cells, are shown (vector-only control [pDISPLAY] as well as human and chicken ACE2). The data underlying this figure may be found in S1 Data and S1 Raw Images. ACE2, angiotensin-converting enzyme 2; BHK-21, baby hamster kidney; MOI, multiplicity of infection; SARS-CoV-2, SARS Coronavirus 2; TCID-50, 50% tissue culture infective dose.
(TIF)

**S7 Fig. Infection of chicken embryo fibroblast DF-1 cell with SARS-CoV-2.** ACE2 expression in transfected DF-1 cells (as indicated) was analysed by flow cytometry (A) or western blot (B). For the flow cytometry, the MFI is shown for chicken and human ACE2-expressing cells (3264 and 9521 cells, respectively). For the controls, the total MFI of the HA labelling is shown as a small number of cells (<10) in the positive gate skewed the data. (C) Similarly, transfected DF-1 cells were infected with SARS-CoV-2 at an MOI of 1, in biological triplicate. Supernatant samples were harvested at 48 hpi for titration by TCID-50. The detection limit for the TCID-50 (DL) is indicated. (D) The mRNA expression levels of 3 proteases relevant for SARS-CoV-2 entry, Furin, Cathepsin B, and TMPRSS2, were evaluated in DF-1 cells by analysis of the publicly available gene expression database series GSE138648 ([18], dataset and methodology (GSM4114984) available at GEO; https://www.ncbi.nlm.nih.gov/geo/). Samples from both uninfected [Control] and ILTV DF-1 cells (4 biological replicates [Rep′] per condition) were examined and FPKM were calculated. The data underlying this figure may be found in S1 Data and S1 Raw Images. ACE2, angiotensin-converting enzyme 2; DF-1, chicken embryonic fibroblast cell line; DL, detection limit, FPKM, fragments per kilobase of exon per

million reads mapped; HA, human influenza hemagglutinin tag; ILTV, infectious laryngotra-cheitis virus-infected; MFI, mean fluorescent intensity; MOI, multiplicity of infection; SARS-CoV-2, SARS Coronavirus 2; TCID-50, 50% tissue culture infective dose; TMPRSS2, trans-membrane protease serine 2.
(TIF)

**S8 Fig. RaTG13 receptor screening.** (A–C) Pseudotype (A) and cell–cell fusion assays (B and C) were established for RaTG13 as described for SARS-CoV-2 and SARS-CoV (see S2 Fig). (D and E) Receptor screening experiments were performed as described for SARS-CoV-2 and SARS-CoV (see S4 Fig). A representative data set is shown from a RaTG13 pseudotype and cell–cell fusion screen, with each screen eventually being performed in triplicate. (F) An XYZ scatter plot examining the receptor usage pattern of SARS-CoV-2 (SARS2; x), SARS-CoV (SARS1; z), and RaTG13 (y) Spike pseudotypes. The mean RLU data from 3 independent experiments is plotted with the human ACE2 value highlighted in red. (G) A Pearson correlation matrix for SARS-CoV-2, SARS-CoV, and RaTG13. The same raw RLU receptor usage values plotted in F were used for these calculations, with pDISPLAY values removed prior to calculation. The data underlying this figure may be found in S1 Data. ACE2, angiotensin-converting enzyme 2; RLU, relative light units; SARS-CoV, SARS Coronavirus; SARS-CoV-2, SARS Coronavirus 2.
(TIF)

**S1 Table. Cell lines utilised in this study to quantify ACE2 mRNA levels and to assess virus permissibility.**
(DOCX)

**S2 Table. Codon-optimised ACE2-expression plasmids used in this study for receptor usage screens.**
(DOCX)

**S3 Table. β-coronavirus glycoproteins used in this study for receptor usage screens.**
(DOCX)

**S4 Table. SARS-CoV-2 and RaTG13 glycoproteins with amino acid mutations in the RBD used in this study to generate chimaeras.**
(DOCX)

**S5 Table. qPCR primer sets used in this study to quantify ACE2 mRNA levels.**
(DOCX)

**S1 Data. Underlying primary data gathered in this study.**
(XLSX)

**S1 Raw Images. Unedited western blot data in this study.**
(PDF)

## Acknowledgments

We would like to thank the following for assistance in establishing the SARS-CoV pseudotype systems: Ed Wright (Viral Pseudotype Unit, University of Sussex), Nigel Temperton and Simon Scott (Viral Pseudotype Unit, University of Kent), Brian Willett (University of Glasgow Centre for Virus Research), Emma Bentley and Giada Mattiuzzo (National Institute for Biological Standards and Control [NIBSC]) and Michael Letko (National Institute of Allergy and Infectious Diseases). We also acknowledge the support of Nadine Lewis (BioBasic) for help

with gene synthesis as well as The Pirbright Institute's Flow Cytometry Facility and The Pirbright Institute Cell Servicing Unit.

## Author Contributions

**Conceptualization:** Helena J. Maier, Erica Bickerton, Stephen C. Graham, Dalan Bailey.

**Data curation:** Dalan Bailey.

**Formal analysis:** Carina Conceicao, Nazia Thakur, Dalan Bailey.

**Funding acquisition:** Dalan Bailey.

**Investigation:** Carina Conceicao, Nazia Thakur, Stacey Human, James T. Kelly, Leanne Logan, Dagmara Bialy, Sushant Bhat, Phoebe Stevenson-Leggett, Adrian K. Zagrajek, Philippa Hollinghurst, Michal Varga, Christina Tsirigoti, Matthew Tully, Chris Chiu, Katy Moffat, Adrian Paul Silesian, John A. Hammond, Holly Shelton, Isabelle Dietrich, Stephen C. Graham, Dalan Bailey.

**Methodology:** Carina Conceicao, Nazia Thakur, John A. Hammond, Stephen C. Graham, Dalan Bailey.

**Project administration:** Carina Conceicao, Nazia Thakur, Dalan Bailey.

**Supervision:** Dalan Bailey.

**Visualization:** Stephen C. Graham, Dalan Bailey.

**Writing – original draft:** Carina Conceicao, Nazia Thakur, Dalan Bailey.

**Writing – review & editing:** Carina Conceicao, Nazia Thakur, John A. Hammond, Erica Bickerton, Stephen C. Graham, Dalan Bailey.

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
