## [Editor Report · Decision Letter 0]

18 Aug 2020

Dear Dr Bailey, 

Thank you for submitting your manuscript entitled "The SARS-CoV-2 Spike protein has a broad tropism for mammalian ACE2 proteins" for consideration as a Research Article by PLOS Biology. Sorry for the delay incurred during this busy holiday period.

Your manuscript has now been evaluated by the PLOS Biology editorial staff, and I'm writing to let you know that we would like to send your submission out for external peer review.

Please re-submit your manuscript within two working days, i.e. by Aug 20 2020 11:59PM.

Kind regards,

Roli Roberts

Senior Editor

PLOS Biology

---

## [Decision Letter · Decision Letter 1]

5 Oct 2020

Dear Dalan,

Thank you very much for submitting your manuscript "The SARS-CoV-2 Spike protein has a broad tropism for mammalian ACE2 proteins" for consideration as a Research Article at PLOS Biology. Your manuscript has been evaluated by the PLOS Biology editors, an Academic Editor with relevant expertise, and by two independent reviewers. We also recruited a third reviewer, but they have failed to submit in timely fashion, so we proceeded to a decision without them. If they send comments then we'll forward them.

IMPORTANT:

a) You'll see that the reviewers are both broadly positive about your study, but both note the existence of several prior studies that they feel might compromise the novelty of your findings. However, we have a strong anti-scooping policy (https://journals.plos.org/plosbiology/article?id=10.1371/journal.pbio.2005203) – according to this, when considering novelty, we ignore all competing papers published during the 6 months running up to submission. The earliest publication date of the four papers mentioned by reviewer #1 is March 2020; reviewer #2 mentions one of these plus an unpublished arXiv preprint. Your paper was submitted on Aug 7th, so well within the 6-month scooping policy deadline of even the original Zhou Nature paper. We therefore did not consider those papers as impacting the advance of your study...

b) ...however, the Academic Editor has asked me to emphasise that you must cite all of these papers, and compare/contrast your findings. We see such comparisons as a significant added value that our anti-scooping policy brings.

c) Please attend to all of the other concerns raised by the reviewers.

d) As part of a commitment between PLOS (and other publishers) and the World Health Organisation to ensure that all relevant clinical information about this outbreak is shared quickly, we will be notifying the WHO directly about your study/manuscript. We strongly encourage you to post your manuscript as a preprint if you have not already done so. Please see here for the statement coordinated by the Wellcome Trust: https://wellcome.ac.uk/press-release/sharing-research-data-and-findings-relevant-novel-coronavirus-covid-19-outbreak.

In light of the reviews (below), we are pleased to offer you the opportunity to address the comments from the reviewers in a revised version that we anticipate should not take you very long. We will then assess your revised manuscript and your response to the reviewers' comments and we may consult the reviewers again.

We expect to receive your revised manuscript within 1 month.

**IMPORTANT - SUBMITTING YOUR REVISION**

*Resubmission Checklist*

*Published Peer Review*

*PLOS Data Policy*

*Blot and Gel Data Policy*

Sincerely,

Roli

Senior Editor,

rroberts@plos.org,

PLOS Biology

REVIEWERS' COMMENTS:

Reviewer #1:

PBIOLOGY-D-20-02408

The SARS-CoV-2 Spike protein has a broad tropism for mammalian ACE2 proteins by Conceicao et al examines a diverse array of 22 non-human ACE2 proteins for their ability to allow SARS-CoV-2 binding and infection. These ACE2 proteins have range of 76-99% amino acid identity relative to human ACE2 and include livestock, domesticated mammals and avian species. Given the ongoing pandemic, with well over a million new cases per week, the concern of reverse zoonosis is real and already been demonstrated in several countries. A greater understanding of at-risk species will help with disease surveillance, and potentially improve our scientific understanding of SARS2 by highlighting potential animal models for further study. This work is technically very sound, although only six of the ACE2 proteins investigated were assessed in the context of live SARS-CoV-2 infection. The breadth of SARS-CoV-2 ACE2 usage detailed in this manuscript is not surprising given the range of small animal models being used in laboratory experiments and the well-publicized cases of mink, tiger, domestic cat, and domestic dog infection. The comparison with SARS-CoV receptor usage in the pseudovirus and cell-cell fusion assays is a strength, as is the mutational analysis comparing the spikes of SARS2 and RaTG13. This work has extensive overlap with recently published papers by Schlottau (Lancet), Zhao (JV), and Shi (Science) and even Zhou (Nature, initial characterization paper). 

Minor Points

Amino acid 498 has also been changes in recent mouse adaptation work, it might be useful to comment on the flexibility/requirement for this contact residue in infection.

Fig 5E - The chimera proteins don't appear to have cleavage products. Any reason or speculation as to why that would be?

The discussion is very long and detailed, the text associated with figures 5 and 6 would be better suited to the results section.

Major Points

While some of these species have not been experimentally tested so far, at this point it is well demonstrated through naturally infections that many species are vulnerable to SARS2 infection - cats, dogs, tigers and mink have all been naturally infected. Hamsters have been used as a small animal model of infection in several publications.

Previous work has shown that pigs and pig cells do not get infected while ferrets do get infected, how do the authors explain these discrepancies with their results?

Figure 3D - What proteases are present in the DF-1 cells? More detail is needed for this to be a convincing negative result.

What cell types from each of the species in Figure 3A were infected? Was there reason to believe these cells should be infectable (ie gastrointestinal or airway cells)? Similarly, DF-1 cells are chicken cells but what type of chicken cell, do they express the needed proteases?

Reviewer #2:

SARS-CoV-2 emergence may have occurred via an intermediate host between bats and humans. Coronavirus (CoV) entry plays an important role in CoV cross-species transmission. The Authors seek to understand the molecular interactions between SARS-CoV-2 Spike (S) protein and ACE2 proteins from a variety of vertebrate sources to investigate tropism and possibly inform choices on animals for SARS-CoV-2 research. They use cell-cell fusion, pseudotype virus and, importantly, actual SARS-CoV-2 virus replication assays to examine SARS-CoV-2 S interaction with ACE2 orthologues. Results obtained were compared to RaTG13 S interaction with ACE2 orthologues to provide context using the closest known sequence isolate to SARS-CoV-2.

In general, the manuscript is well written and experiments well designed and controlled. There have been two other manuscripts published or available looking at some similar questions (ref 28 and 29 [preprint] in the manuscript as mentioned by the Authors) but this manuscript has the most thorough and interesting examination of SARS-CoV-2, SARS-CoV and RaTG13 Spike/ACE2 interactions to date.

Because of the obviously heightened interest in SARS-CoV-2 during the pandemic and the likely interest in CoV tropism in the future, researchers from many disciplines will find this manuscript interesting.

Comments

1. In Fig1C, the ACE2 ectodomain map lists blue (divergent) to purple (conserved) residues. Clarification on what type of residue difference is considered divergent vs what type is considered conserved would be helpful to fully understand that figure.

2. Fig2A. Heat map. It is unclear what the error or deviation from the mean is for the values listed here. Standard deviation or standard error need not be included in the heat map for every value but an error range could be listed in the legend (something like "standard deviation for the means range from 0.3-15%" for instance).

3. Sup Fig 6D. The chicken DF-1 cell line data is interesting. Overexpression of chicken ACE2 did not rectify lack of SARS-CoV-2 pseudotype entry. An interesting question results, is this due to some post-receptor binding factor or lack thereof? Why was human ACE2 not included in this expt to potentially address some aspects of this question? 

A further small concern with these expts is the low transfection efficiency (12%) for the DF-1 cells. Could this contribute to relative resistance to infection in DF-1 overexpression expt? Transfection efficiency for Pig ACE2 in BHK-21 cells was 20% and yielded robust virus entry so not sure if efficiency is an issue.

Minor comments.

1. Sup Fig 6C. BHK-21 cell graph labeled as BHK-1?

2. Line 488. "syncytia formation" should be replaced by "syncytium formation". The latter term has historically been accepted and is most likely to be correct.

3. When the Fig 1 guinea pig data is discussed in the results section, it would be helpful if there was a "(see Methods)" note or some other info to convey the details of the possible sequence issues on lines 702-4.

---

## [Editor Report · Decision Letter 2]

3 Nov 2020

Dear Dalan,

Thank you for submitting your revised Research Article entitled "The SARS-CoV-2 Spike protein has a broad tropism for mammalian ACE2 proteins" for publication in PLOS Biology. In order to avoid a further round of review, the Academic Editor kindly agreed to check your responses and revisions.

Based on the Academic Editor's assessment, we're delighted to let you know that we're now editorially satisfied with your manuscript. However before we can formally accept your paper and consider it "in press", we also need to ensure that your article conforms to our guidelines. A member of our team will be in touch shortly with a set of requests. As we can't proceed until these requirements are met, your swift response will help prevent delays to publication. Please also make sure to address the data and other policy-related requests noted at the end of this email.

IMPORTANT:

a) Many thanks for complying with our Data Policy by supplying the underlying data as “Raw data for graphs.xlsx.” Please could you re-name it "S1_Data. xlsx" and cite it in all relevant main and supplementary Figure legends, e.g. "The data underlying this Figure may be found in S1 Data."

b) Please change the "S" in "Spike" in the title to lower case.

- a cover letter that should detail your responses to any editorial requests, if applicable

*Copyediting*

*Published Peer Review History*

*Early Version*

Sincerely,

Roli

Senior Editor,

rroberts@plos.org,

PLOS Biology

---

## [Editor Report · Decision Letter 3]

17 Nov 2020

Dear Dr Bailey,

On behalf of my colleagues and the Academic Editor, Bill Sugden, I am pleased to inform you that we will be delighted to publish your Research Article in PLOS Biology. 

PRODUCTION PROCESS

Before publication you will see the copyedited word document (within 5 business days) and a PDF proof shortly after that. The copyeditor will be in touch shortly before sending you the copyedited Word document. We will make some revisions at copyediting stage to conform to our general style, and for clarification. When you receive this version you should check and revise it very carefully, including figures, tables, references, and supporting information, because corrections at the next stage (proofs) will be strictly limited to (1) errors in author names or affiliations, (2) errors of scientific fact that would cause misunderstandings to readers, and (3) printer's (introduced) errors. Please return the copyedited file within 2 business days in order to ensure timely delivery of the PDF proof. 

If you are likely to be away when either this document or the proof is sent, please ensure we have contact information of a second person, as we will need you to respond quickly at each point. Given the disruptions resulting from the ongoing COVID-19 pandemic, there may be delays in the production process. We apologise in advance for any inconvenience caused and will do our best to minimize impact as far as possible.

EARLY VERSION

PRESS 

Kind regards,

Vita Usova

Publication Assistant, 

PLOS Biology

on behalf of

Roland Roberts,

Senior Editor

PLOS Biology